# A Recursive Decomposition Framework for Causal Structure Learning in the Presence of Latent Variables

Zheng Li [1 2 3]   Feng Xie [2]   Shenglan Nie [2]   Xichen Guo [2]   Ruxin Wang [1]   Hao Zhang [1]

## Abstract

Constraint-based causal discovery is widely used for learning causal structures, but heavy reliance on conditional independence (CI) testing makes it computationally expensive in high-dimensional settings. To mitigate this limitation, many divide-and-conquer frameworks have been proposed, but most assume causal sufficiency, i.e., no latent variables. In this paper, we show that divide-and-conquer strategies can be theoretically generalized beyond causal sufficiency to settings with latent variables. Specifically, we propose a recursive decomposition framework, termed DICOLA, that enables divide-and-conquer causal discovery in the presence of latent variables. It recursively decomposes the global learning task into smaller subproblems and integrates their solutions through a principled reconstruction step to recover the global structure. We theoretically establish the soundness and completeness of the proposed framework. Extensive experiments on synthetic data demonstrate that our approach significantly improves computational efficiency across a range of causal discovery algorithms, while experiments on a real-world dataset further illustrate its practical effectiveness. Our source code is available at github.com/zhengli0060/DiCoLa-ICML2026.

## 1. Introduction

Causal discovery, i.e., inferring causal relations from data, is a fundamental problem across many disciplines, including computer science (Jonas et al., 2017; Pearl, 2018; Schölkopf, 2022), social science (Spirtes et al., 2000), epidemiology (Hernán & Robins, 2010), biology (Glymour et al., 2019), and neuroscience (Smith et al., 2011; Sanchez-Romero et al., 2019). The resulting causal relationships are essential for predicting how a system would respond to external interventions, which is central to both understanding and manipulating complex systems (Pearl, 2009). For instance, in medical diagnosis, understanding the causal relationships between symptoms and diseases enables clinicians to identify root causes more efficiently and deliver more targeted interventions. However, learning such relationships from purely observational data is inherently challenging, and the presence of latent variables further exacerbates this difficulty by inducing complex dependencies among observed variables (Spirtes & Zhang, 2016).

The seminal FCI (Fast Causal Inference) algorithm, proposed by Spirtes et al. (1995), can recover causal structures in the presence of latent variables by leveraging conditional independence (CI) relations derived from observational data. Subsequent work has focused on improving its efficiency by reducing the number of required CI tests. For example, RFCI (Colombo et al., 2012) accelerates learning by relaxing certain CI search steps, making it more suitable for sparse, high-dimensional systems. Rohekar et al. (2021) proposed the ICD algorithm, which restricts CI conditioning sets based on the topological distance between tested variables, thereby reducing the number of CI tests. Additionally, Akbari et al. (2021) introduced L-MARVEL, which identifies and recursively removes a specific class of variables to further reduce CI testing complexity. Other related developments include (Pellet & Elisseeff, 2008a; Claassen & Heskes, 2011; Claassen et al., 2013; Mokhtarian et al., 2023; 2025). Despite these algorithmic improvements, the computational cost of CI testing remains prohibitively high, especially in large-scale systems.

Divide-and-conquer frameworks provide an effective strategy for reducing the cost of CI testing, particularly in high-dimensional settings. A series of such approaches (Xie et al., 2006; Xie & Geng, 2008; Cai et al., 2013; 2017a) and their variants (Liu et al., 2017; Cai et al., 2017b; Zhang et al., 2019; 2020; Rahman et al., 2021; Zhang et al., 2024a) have been developed. At a high level, as illustrated in Figure 1, these methods decompose the global causal structure

[1] Shenzhen Institutes of Advanced Technology, Chinese Academy of Sciences, Shenzhen, China [2] Department of Applied Statistics, Beijing Technology and Business University, Beijing, China [3] College of Computer Science and Artificial Intelligence, Fudan University, Shanghai, China. Correspondence to: Feng Xie <fengxie@btbu.edu.cn>, Ruxin Wang <rx.wang@siat.ac.cn>, Hao Zhang <h.zhang10@siat.ac.cn>.

*Proceedings of the 43rd International Conference on Machine Learning*, Seoul, South Korea. PMLR 306, 2026. Copyright 2026 by the author(s).

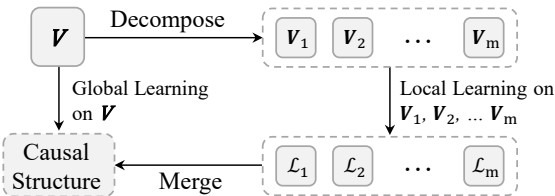

*Figure 1.* The divide-and-conquer for the full variable set $\mathbf{V}$. learning task over the full variable set $\mathbf{V}$ into subproblems defined on smaller subsets $\mathbf{V}_1, \mathbf{V}_2, \ldots, \mathbf{V}_m$. Each subproblem is solved independently using existing learners such as IC (Verma & Pearl, 1990) or PC (Spirtes & Glymour, 1991), and the resulting substructures are then merged into a global solution. By replacing exhaustive CI testing over $\mathbf{V}$ with localized tests on subsets $\mathbf{V}_i$, these methods significantly shrink the search space and avoid redundant computations (Xie & Geng, 2008; Zhang et al., 2024a). However, all existing divide-and-conquer approaches rely critically on the assumption of *causal sufficiency*—that no latent variable is a common cause of two or more observed variables—which fundamentally limits their applicability in real-world systems where latent variables are often present.

This naturally leads to an important theoretical question:

> *Can divide-and-conquer strategies for causal structure learning be theoretically generalized beyond causal sufficiency to settings with latent variables?*

In this paper, inspired by the recursive divide-and-conquer framework of Xie & Geng (2008) under causal sufficiency, we provide a positive answer to this question. Our main contributions are summarized as follows:

- We establish the first theoretical foundation for divide-and-conquer causal structure learning in the presence of latent variables. Notably, our theoretical results (Theorems 1 to 3) naturally reduce to those of Xie & Geng (2008) under causal sufficiency.

- We propose a novel recursive decomposition framework, termed D$\textsc{i}$C$\textsc{o}$L$\textsc{a}$, which improves the computational efficiency of causal structure learning in the presence of latent variables. We further prove that D$\textsc{i}$C$\textsc{o}$L$\textsc{a}$ is sound and complete.

- We conduct extensive experiments on random and benchmark structures, showing that our framework substantially accelerates FCI (Spirtes et al., 1995), RFCI (Colombo et al., 2012), FCI$^+$ (Claassen et al., 2013), L-MARVEL (Akbari et al., 2021), and ICD (Rohekar et al., 2021). Experiments on real-world datasets further demonstrate its practical effectiveness.

## 2. Related Work

We briefly review constraint-based causal structure learning methods most relevant to our work. For comprehensive surveys, we refer the reader to Spirtes & Zhang (2016); Heinze-Deml et al. (2018); Glymour et al. (2019); Kitson et al. (2023); Mokhtarian et al. (2025).

**In the absence of latent variables.** Classical constraint-based methods such as the IC algorithm (Verma & Pearl, 1990) and the PC algorithm (Spirtes & Glymour, 1991) recover causal structures by testing conditional independences. A number of extensions have been proposed to improve efficiency and robustness (Harris & Drton, 2013; Colombo et al., 2014; Le et al., 2016; Rohekar et al., 2018; Mokhtarian et al., 2021; 2022; Shiragur et al., 2024; Zhang et al., 2024b). To address scalability in high-dimensional systems, a significant line of work develops divide-and-conquer frameworks (Xie et al., 2006; Xie & Geng, 2008; Cai et al., 2013; 2017a), which decompose the global learning problem into smaller subproblems. Subsequent variants further refine the decomposition strategy to improve efficiency and accuracy (Liu et al., 2017; Cai et al., 2017b; Zhang et al., 2019; 2020; Rahman et al., 2021; Zhang et al., 2024a). However, all these methods rely on the assumption of causal sufficiency, i.e., that no latent variable acts as a common cause of multiple observed variables—an assumption not required by our framework.

**In the presence of latent variables.** When latent confounders may be present, the seminal FCI algorithm (Spirtes et al., 1995; Zhang, 2008b) extends PC to this setting and is sound and complete under standard assumptions. Building on FCI, several algorithms aim to improve efficiency. For example, RFCI (Colombo et al., 2012) reduces the number of CI tests by relaxing certain search steps, making it more suitable for high-dimensional sparse graphs. Other developments include logical, hybrid, and local approaches (Pellet & Elisseeff, 2008a; Claassen & Heskes, 2011; Claassen et al., 2013; Ogarrio et al., 2016; Tsirlis et al., 2018; Chen et al., 2023), as well as more recent iterative and recursive techniques (Rohekar et al., 2021; 2023; Akbari et al., 2021; Mokhtarian et al., 2025). While these algorithms can handle latent variables, they still suffer from the high computational cost of CI testing, particularly in high-dimensional settings, as also reflected in the number of CI tests reported in our experiments.

**Gap to our work.** To the best of our knowledge, no prior work has established a theoretically justified divide-and-conquer framework for causal structure learning in the presence of latent confounders.

## 3. Preliminaries

### 3.1. Terminology

We adopt standard terminology from causal graphical models with latent variables (Pearl, 2009; Richardson & Spirtes, 2002; Zhang, 2008a). Individual variables (vertices) are

denoted by uppercase letters (e.g., $V$), and sets of variables by bold uppercase letters (e.g., $\mathbf{V}$). We use standard notions of adjacency, parents, children, spouses, ancestors, descendants, and colliders. Below we introduce several frequently used definitions; additional terminology and formal definitions are provided in Appendix A, and a summary of the main symbols is provided in Table 1.

**Mixed graph.** A *directed mixed graph* contains directed ($\rightarrow$) and bi-directed ($\leftrightarrow$) edges. An *undirected graph* contains only undirected edges ($-$). A *directed path* from $X$ to $Y$ is a path composed of directed edges pointing towards $Y$, i.e., $X \rightarrow \ldots \rightarrow Y$. A non-endpoint vertex $V$ on a path is a *collider* if the edges preceding and succeeding $V$ on the path both have arrowheads into $V$; otherwise, it is a *non-collider*. A path is a *collider path* if every non-endpoint vertex on it is a collider along the path, e.g. , $X \rightarrow A \leftrightarrow \ldots \leftrightarrow B \leftarrow Y$. Two vertices $X$ and $Y$ are *collider connected* if there exists a collider path between $X$ and $Y$, including the case where $X$ and $Y$ are adjacent. An *almost directed cycle* occurs when $X$ is a spouse of $Y$, and there exists a directed path from $X$ to $Y$. A *directed cycle* occurs when $X$ is a child of $Y$, and there exists a directed path from $X$ to $Y$.

**M-separation.** In a directed mixed graph, a path between vertices $X$ and $Y$ is said to be *m-connecting* (active) relative to a set of vertices $\mathbf{Z}$ ($X, Y \notin \mathbf{Z}$) if (i) every non-collider on the path is not in $\mathbf{Z}$, and (ii) every collider on the path has a descendant in $\mathbf{Z}$. Vertices $\mathbf{X}$ and $\mathbf{Y}$ are *m-separated* by $\mathbf{Z}$ if there is no m-connecting path between any $X \in \mathbf{X}$ and $Y \in \mathbf{Y}$ relative to $\mathbf{Z}$, and $\mathbf{Z}$ called a *separating set* for $\mathbf{X}$ and $\mathbf{Y}$. The notion of m-separation generalizes the classical d-separation criterion for DAGs to directed mixed graphs. In an undirected graph, vertices $\mathbf{X}$ are said to be *separated* from vertices $\mathbf{Y}$ by a set of vertices $\mathbf{Z}$ if every path between $\mathbf{X}$ and $\mathbf{Y}$ contains at least one vertex in $\mathbf{Z}$.

**Ancestral Graph.** A directed mixed graph is *ancestral* if it doesn't contain a directed or almost directed cycle. An ancestral graph is called *maximal* (MAG, denoted by $\mathcal{M}$) if for any two non-adjacent vertices, there exists a set of vertices that m-separates them. A MAG is a *directed acyclic graph* (DAG) if it has only directed edges. Two MAGs are *Markov equivalent* if they share the same m-separations. A class of Markov equivalent MAGs $[\mathcal{M}]$ can be represented as a *partially ancestral graph* (PAG), where a tail '$-$' or arrowhead '$>$' occurs if the corresponding mark is tail or arrowhead in every $\mathcal{M} \in [\mathcal{M}]$, and a circle '$\circ$' occurs if the corresponding mark is not invariant in every $\mathcal{M} \in [\mathcal{M}]$.

**Local skeleton.** Given a MAG $\mathcal{M}$ and a vertex subset $\mathbf{K}$, the *local skeleton* over $\mathbf{K}$, denoted by $\mathcal{L}_{\mathbf{K}}$, is the undirected graph that connects $X$ and $Y$ if and only if no subset of $\mathbf{K}$ m-separates $X$ and $Y$ in $\mathcal{G}$. When $\mathbf{K} = \mathbf{O}$, the local skeleton coincides with the skeleton of $\mathcal{M}$. A *tripartition* of a set $\mathbf{K}$ is a collection $(\mathbf{A}, \mathbf{B}, \mathbf{C})$ of three pairwise disjoint subsets whose union is $\mathbf{K}$.

**Conditional independence.** For three disjoint variable sets $\mathbf{X}$, $\mathbf{Y}$, and $\mathbf{Z}$, we write $\mathbf{X} \perp\!\!\!\perp \mathbf{Y} \mid \mathbf{Z}$ to denote that $\mathbf{X}$ is statistically independent of $\mathbf{Y}$ given $\mathbf{Z}$, and $\mathbf{X} \not\perp\!\!\!\perp \mathbf{Y} \mid \mathbf{Z}$ otherwise.

**Markov blanket.** Given a variable set $\mathbf{V}$, the *Markov blanket (MB)* of a variable $X \in \mathbf{V}$, denoted by $\mathrm{MB}_{\mathbf{V}}(X)$, is the minimal subset of $\mathbf{V} \setminus \{X\}$ such that $X$ is conditionally independent of all other variables given $\mathrm{MB}_{\mathbf{V}}(X)$.

### 3.2. Problem Setup

We consider a causal graphical model in which the full variable set $\mathbf{V}$ consists of observed variables $\mathbf{O}$ and latent variables $\mathbf{L}$. The underlying causal structure is represented by a DAG over $\mathbf{V}$, and we assume no selection bias. We further assume the standard causal Markov and Faithfulness conditions, under which conditional independences in the observed distribution are equivalent to m-separations in the underlying structure (Pearl, 2009; Zhang, 2008a).

**Goal.** Given data over $\mathbf{O}$, our goal is to develop a divide-and-conquer framework that improves the computational efficiency of causal structure learning while correctly recovering the partial ancestral graph (PAG) representing the Markov equivalence class of the underlying MAG.

## 4. Foundations of Recursive Decomposition

In this section, we establish the theoretical foundations that justify divide-and-conquer causal structure learning in the presence of latent variables. Specifically, we address the following fundamental question:

*Under what structural conditions can global causal structure learning be decomposed into smaller subproblems while preserving m-separation relations?*

Before formalizing the answer, we first outline the recursive decomposition procedure. Intuitively, given a set of observed variables $\mathbf{O}$, the procedure identifies a tripartition $(\mathbf{A}, \mathbf{B}, \mathbf{C})$ such that $\mathbf{A} \perp\!\!\!\perp \mathbf{B} \mid \mathbf{C}$ (we describe how to identify such tripartitions in detail in Section 5.1). The global learning problem on $\mathbf{O}$ is then decomposed into two smaller subproblems defined on $\mathbf{A} \cup \mathbf{C}$ and $\mathbf{B} \cup \mathbf{C}$. This process is applied recursively until no further decomposition is possible, yielding a hierarchical structure that can be represented as a binary *decomposition tree* (see Figure 2(d)).

We now answer the fundamental question by establishing the following results on the behavior of m-separation in a MAG (Theorems 1 and 2). These results reveal a form of *top-down transitivity* of m-separation relations across the decomposition tree.

**Theorem 1.** *Let $\mathbf{A}$, $\mathbf{B}$, and $\mathbf{C}$ be three disjoint subsets of*

*vertices in a MAG $\mathcal{M}$ such that $\mathbf{A} \perp\!\!\!\perp \mathbf{B} \mid \mathbf{C}$. For any $X \in \mathbf{A}$ and $Y \in \mathbf{A} \cup \mathbf{C}$, $X$ and $Y$ are m-separated by a subset of $\mathbf{A} \cup \mathbf{B} \cup \mathbf{C}$ if and only if they are m-separated by a subset of $\mathbf{A} \cup \mathbf{C}$.*

Theorem 1 formalizes the intuition that, once a valid tripartition $(\mathbf{A}, \mathbf{B}, \mathbf{C})$ is identified with $\mathbf{A} \perp\!\!\!\perp \mathbf{B} \mid \mathbf{C}$, variables in $\mathbf{B}$ become irrelevant when determining separating sets for pairs $(X, Y)$ with $X \in \mathbf{A}$ and $Y \in \mathbf{A} \cup \mathbf{C}$. If a separating set exists in the full set $\mathbf{O}$, one must also exist within $\mathbf{A} \cup \mathbf{C}$, and conversely.

**Theorem 2.** *Let $\mathbf{A}$, $\mathbf{B}$, and $\mathbf{C}$ be three disjoint subsets of vertices in a MAG $\mathcal{M}$ such that $\mathbf{A} \perp\!\!\!\perp \mathbf{B} \mid \mathbf{C}$. For any distinct vertices $X, Y \in \mathbf{C}$, they are m-separated by a subset of $\mathbf{A} \cup \mathbf{B} \cup \mathbf{C}$ if and only if they are m-separated by a subset of $\mathbf{A} \cup \mathbf{C}$ or by a subset of $\mathbf{B} \cup \mathbf{C}$.*

Theorem 2 addresses pairs of vertices within $\mathbf{C}$. For any distinct $X, Y \in \mathbf{C}$, if a separating set exists in the full variable set $\mathbf{O}$, then one can always be found within either $\mathbf{A} \cup \mathbf{C}$ or $\mathbf{B} \cup \mathbf{C}$. Conversely, if neither subset contains a separating set, then $X$ and $Y$ remain m-connected given any subset of $\mathbf{O}$.

**Remark 1.** *It is worth emphasizing that if $\mathbf{A} \cup \mathbf{B} \cup \mathbf{C} \subseteq \mathbf{O}$, the conclusions of Theorems 1 and 2 still hold. This flexibility enables the recursive application of the decomposition scheme while preserving the transitivity of m-separation relations.*

Based on Theorems 1 and 2, we obtain a unified characterization for any subset $\mathbf{K} \subseteq \mathbf{O}$ that admits a tripartition $(\mathbf{A}, \mathbf{B}, \mathbf{C})$ with $\mathbf{A} \perp\!\!\!\perp \mathbf{B} \mid \mathbf{C}$: for any distinct variables $X, Y \in \mathbf{K}$, the pair $(X, Y)$ is m-separated by a subset of $\mathbf{K}$ if and only if it is m-separated by a subset of $\mathbf{A} \cup \mathbf{C}$ or $\mathbf{B} \cup \mathbf{C}$. This characterization directly leads to a reconstruction principle: the skeleton of the MAG over $\mathbf{O}$ can be recovered by recursively merging the local skeletons obtained from its subproblems, which we formalize in the following theorem.

**Theorem 3.** *Let $\mathbf{A}$, $\mathbf{B}$, and $\mathbf{C}$ be three disjoint subsets of $\mathbf{O}$ such that $\mathbf{A} \perp\!\!\!\perp \mathbf{B} \mid \mathbf{C}$, and let $\mathbf{K} = \mathbf{A} \cup \mathbf{B} \cup \mathbf{C}$. Let $\mathcal{L}_{\mathbf{K}}$ denote the local skeleton over $\mathbf{K}$, and let $\mathcal{L}_{\mathbf{A} \cup \mathbf{C}}$ and $\mathcal{L}_{\mathbf{B} \cup \mathbf{C}}$ denote the local skeletons over $\mathbf{A} \cup \mathbf{C}$ and $\mathbf{B} \cup \mathbf{C}$, respectively. Then, the local skeleton $\mathcal{L}_{\mathbf{K}}$ can be constructed by merging $\mathcal{L}_{\mathbf{A} \cup \mathbf{C}}$ and $\mathcal{L}_{\mathbf{B} \cup \mathbf{C}}$ through the following steps:*

*Step 1: Initialize the edge set of $\mathcal{L}_{\mathbf{K}}$ as the union of the edge sets of $\mathcal{L}_{\mathbf{A} \cup \mathbf{C}}$ and $\mathcal{L}_{\mathbf{B} \cup \mathbf{C}}$.*

*Step 2: For any pair of distinct vertices $X, Y \in \mathbf{C}$, remove the edge $\langle X, Y \rangle$ from $\mathcal{L}_{\mathbf{K}}$ if it is absent from either $\mathcal{L}_{\mathbf{A} \cup \mathbf{C}}$ or $\mathcal{L}_{\mathbf{B} \cup \mathbf{C}}$.*

Theorem 3 provides a constructive rule for recovering the global skeleton from the local skeletons of its subproblems. Edges between variables outside the separator set $\mathbf{C}$ are inherited directly from the corresponding subproblems, while

edges within $\mathbf{C}$ are retained only when supported by both sides of the decomposition.

Taken together, the above results (Theorems 1–3) reveal the following intuition. M-separation relations propagate in a top-down manner during decomposition: any separating relation that holds globally can be captured within an appropriate subproblem. Conversely, during the merging phase, m-connection relations exhibit a bottom-up consistency: if two variables cannot be separated in the relevant subproblems, they remain connected in the global structure. This dual behavior underlies the correctness of our recursive decomposition framework.

## 5. Implementation of Recursive Decomposition

In this section, we describe how to operationalize the recursive decomposition framework. We first present a practical method for identifying valid decompositions in Section 5.1, based on structural properties of certain undirected graphs. We then formalize the DICOLA framework in Section 5.2.

### 5.1. Identifying Valid Decompositions

While Section 4 establishes the theoretical foundation for recursive decomposition, its success hinges on the ability to find a *valid decomposition*—that is, a tripartition $(\mathbf{A}, \mathbf{B}, \mathbf{C})$ of a variable set $\mathbf{K}$ satisfying $\mathbf{A} \perp\!\!\!\perp \mathbf{B} \mid \mathbf{C}$. This raises a practical question:

*How can such a tripartition be efficiently constructed from observational data?*

To address this, we introduce the notion of an *undirected independence graph* (UIG), adapted from Lauritzen (1996); Xie & Geng (2008). While their formulation is defined for DAGs under causal sufficiency, we extend this concept to MAGs to accommodate latent variables. The UIG serves as a convenient surrogate structure for identifying valid tripartitions of a given variable set.

**Definition 1 (Undirected Independence Graph).** *Let $\mathcal{M}$ be a MAG over $\mathbf{O}$. An* undirected independence graph (UIG) *of $\mathcal{M}$ relative to a subset $\mathbf{K} \subseteq \mathbf{O}$, denoted by $\overline{\mathcal{G}}_{\mathbf{K}}$, is an undirected graph with vertex set $\mathbf{K}$ such that for any disjoint sets $\mathbf{X}, \mathbf{Y}, \mathbf{Z} \subseteq \mathbf{K}$, if $\mathbf{Z}$ separates $\mathbf{X}$ and $\mathbf{Y}$ in $\overline{\mathcal{G}}_{\mathbf{K}}$, then $\mathbf{X}$ and $\mathbf{Y}$ are m-separated by $\mathbf{Z}$ in $\mathcal{M}$.*

This definition leads to a key observation: vertex separation in the UIG provides a sufficient condition for m-separation in the underlying MAG. Consequently, identifying a valid tripartition $(\mathbf{A}, \mathbf{B}, \mathbf{C})$ reduces to constructing an appropriate UIG $\overline{\mathcal{G}}_{\mathbf{K}}$ of $\mathcal{M}$. To this end, we introduce the *augmented graph*, which we show to be a particular UIG relative to $\mathbf{O}$, as established in Propositions 1 and 2.

**Definition 2 (Augmented Graph).** *Let $\mathcal{M}$ be a MAG. The augmented graph of $\mathcal{M}$, denoted by $(\mathcal{M})^a$, is the undirected*

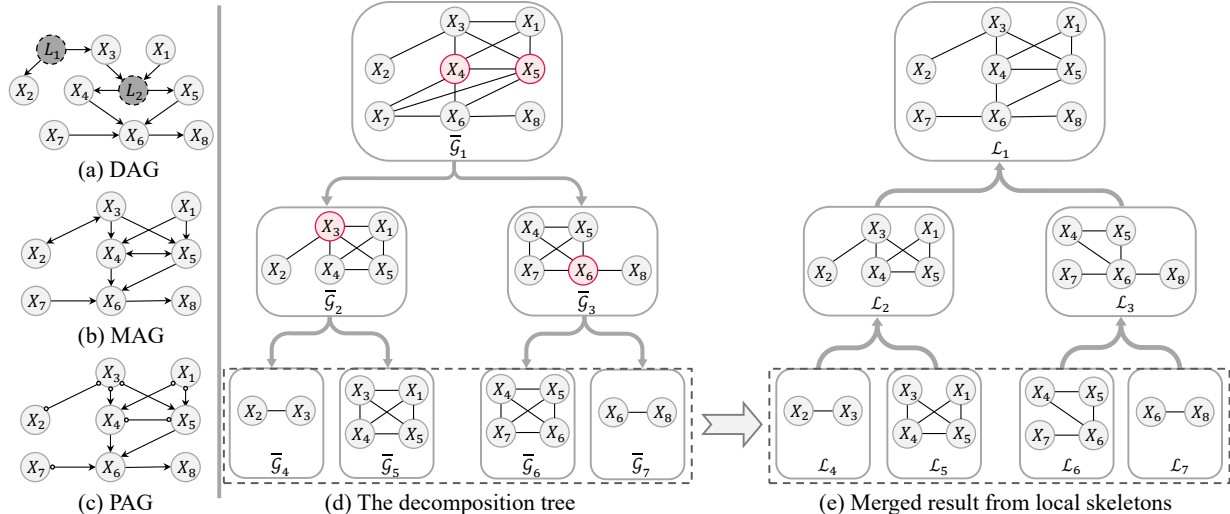

*Figure 2.* (a) An underlying causal structure (adapted from MILDEW network (Jensen & Jensen, 1996)), where $L_1$ and $L_2$ are latent variables. (b) The MAG characterizes the causal relations over the observed variables in (a). (c) The inferred PAG from observed variables. (d) The decomposition tree generated by DICOLA, where each node $\overline{\mathcal{G}}_i$ represents a sub-problem. (e) Merging process of local skeletons.

*graph over the same vertex set as $\mathcal{M}$ in which two vertices are adjacent if and only if they are* collider connected *in $\mathcal{M}$.*

Two vertices $X$ and $Y$ are *collider connected* if there exists a collider path connecting them, including the degenerate case where $X$ and $Y$ are adjacent in $\mathcal{M}$. As illustrated in Figure 2(b), $\overline{\mathcal{G}}_1$ corresponds to the augmented graph of the MAG shown there.

**Proposition 1.** *Let $\mathcal{M}$ be a MAG over $\mathbf{O}$. The augmented graph $(\mathcal{M})^a$ is a UIG of $\mathcal{M}$ relative to $\mathbf{O}$. Moreover, $(\mathcal{M})^a$ is* minimal *in the sense that removing any edge from $(\mathcal{M})^a$ would introduce a separation relation that does not correspond to any m-separation in $\mathcal{M}$.*

Proposition 1 establishes that $(\mathcal{M})^a$ is the sparsest UIG, which exposes more separation opportunities. Intuitively, $(\mathcal{M})^a$ captures the maximum number of admissible decompositions among all possible UIGs relative to $\mathbf{O}$. Importantly, the structure of the augmented graph is intimately linked to the statistical properties of the variables via Markov blankets, as shown below Proposition 2.

**Proposition 2.** *For any two distinct vertices $X, Y \in \mathbf{O}$, $X$ and $Y$ are adjacent in $(\mathcal{M})^a$ if and only if $Y$ belongs to the Markov blanket of $X$ relative to $\mathbf{O}$.*

Proposition 2 yields a direct way to construct the minimal UIG over $\mathbf{O}$: by learning the MB of each variable relative to $\mathbf{O}$, we can reconstruct the augmented graph $(\mathcal{M})^a$. The resulting procedure is summarized in Algorithm 1.

It is worth noting that when $\mathbf{K} = \mathbf{O}$, Algorithm 1 exactly recovers the augmented graph $(\mathcal{M})^a$. Furthermore, when applied to any subset $\mathbf{K} \subset \mathbf{O}$ during recursion, the Markov-blanket-based construction still yields a valid and minimal

---

**Algorithm 1:** CONSTRUCTUIG

**Input:** Observed variable set $\mathbf{K}$
**Output:** The undirected independence graph $\overline{\mathcal{G}}_\mathbf{K}$

1  Initialize $\overline{\mathcal{G}}_\mathbf{K}$ with vertice set $\mathbf{K}$ and no edges;
2  **for** $X \in \mathbf{K}$ **do**
3      ▷ Learn MB of $X$ relative to $\mathbf{K}$
4      $\text{MB}_\mathbf{K}(X) \leftarrow \text{MBLEARN}(\mathbf{K}, X)$;
5      Add edges $\{\langle X, Y \rangle : Y \in \text{MB}_\mathbf{K}(X)\}$ to $\overline{\mathcal{G}}_\mathbf{K}$;
6  **return** $\overline{\mathcal{G}}_\mathbf{K}$

---

UIG, as formalized in Proposition 3.

**Proposition 3.** *Let $\mathcal{M}$ be a MAG over $\mathbf{O}$, and let $\overline{\mathcal{G}}_\mathbf{K}$ be the UIG constructed by Algorithm 1 over $\mathbf{K} \subset \mathbf{O}$. Then, $\overline{\mathcal{G}}_\mathbf{K}$ is an UIG of $\mathcal{M}$ relative to $\mathbf{K}$. Furthermore, $\overline{\mathcal{G}}_\mathbf{K}$ is a minimal UIG of $\mathcal{M}$ relative to $\mathbf{K}$.*

Proposition 3 guarantees that the decomposition procedure remains valid when applied recursively to subproblems. Moreover, $\overline{\mathcal{G}}_\mathbf{K}$ preserves the maximal separation information available within $\mathbf{K}$.

**Remark 2.** *Based on Propositions 1 to 3, a valid tripartition of any subset $\mathbf{K} \subseteq \mathbf{O}$ can be identified through Markov blanket learning. The* FINDDECOMPOSITION *procedure (Algorithm 2) works in two steps. It first calls* CONSTRUCTUIG *to build the minimal UIG $\overline{\mathcal{G}}_\mathbf{K}$. It then applies an undirected graph partitioning method (e.g., junction trees (Jensen & Jensen, 1994)) to find a vertex separator $\mathbf{C}$, which partitions $\mathbf{K}$ into $(\mathbf{A}, \mathbf{B}, \mathbf{C})$ with no edges between $\mathbf{A}$ and $\mathbf{B}$ in the UIG.*

When multiple valid tripartitions are available for a ver-

---

**Algorithm 2:** FINDDECOMPOSITION

**Input:** Observed variable set $\mathbf{K}$
**Output:** $(\mathbf{A}, \mathbf{B}, \mathbf{C}, \textit{flag})$

1   $\overline{\mathcal{G}}_{\mathbf{K}} \leftarrow$ CONSTRUCTUIG($\mathbf{K}$);
2   $(\mathbf{A}, \mathbf{B}, \mathbf{C}) \leftarrow$ VertexCut($\overline{\mathcal{G}}_{\mathbf{K}}$);
3   $\textit{flag} \leftarrow$ whether $\overline{\mathcal{G}}_{\mathbf{K}}$ has a tripartition $(\mathbf{A}, \mathbf{B}, \mathbf{C})$;
4   **return** $(\mathbf{A}, \mathbf{B}, \mathbf{C}, \textit{flag})$

---

tex set $\mathbf{K}$, we follow the selection strategy of Zhang et al. (2024a). In particular, we choose the tripartition that minimizes the following balancing score:

$$\text{score} = \frac{|\mathbf{C}|}{\min\big(|\mathbf{A} \cup \mathbf{C}|, \ |\mathbf{B} \cup \mathbf{C}|\big)}.$$

This criterion favors balanced subproblems while keeping the separator set $\mathbf{C}$ small.

### 5.2. The DiCoLa Framework

The DICOLA framework, formalized in Algorithm 3, follows a systematic "divide-learn-merge" paradigm. It first applies a top-down recursive decomposition to break the global problem into smaller subproblems, then uses a base structure learning algorithm to solve each subproblem, and finally performs a bottom-up merging step to reconstruct the global causal skeleton. For clarity, we illustrate the workflow with an example in Example 1.

**Example 1** (Illustration of DICOLA). Consider the graphs shown in Figure 2. The underlying system contains eight observed variables $\{X_1, \ldots, X_8\}$ and two latent variables $\{L_1, L_2\}$, as illustrated by the ground-truth DAG in Figure 2(a). The corresponding MAG after marginalizing out the latent variables is shown in Figure 2(b). DICOLA proceeds through the following phases to recover the final PAG (Figure 2(c)):

- **Phase 1: Top-down decomposition.** DICOLA starts from the full variable set $\mathbf{O}$. The procedure FINDDECOMPOSITION (Algorithm 2) first calls CONSTRUCTUIG (Algorithm 1) to learn the UIG $\overline{\mathcal{G}}_1$ over $\mathbf{O}$, and then identifies a vertex separator that yields a valid tripartition $(\mathbf{A}, \mathbf{B}, \mathbf{C})$. In this example, $\mathbf{C} = \{X_4, X_5\}$ (highlighted in red in Figure 2(d)) disconnects the graph, producing two subproblems $\{X_1, \ldots, X_5\}$ and $\{X_4, \ldots, X_8\}$. The process continues recursively: $\{X_1, \ldots, X_5\}$ is further decomposed into $\{X_1, X_3, X_4, X_5\}$ and $\{X_2, X_3\}$ using $X_3$ as the separator, while $\{X_4, \ldots, X_8\}$ is split into $\{X_4, X_5, X_6, X_7\}$ and $\{X_6, X_8\}$ via $X_6$. This produces the decomposition tree shown in Figure 2(d).
- **Phase 2: Local skeleton learning.** Once the decomposition reaches leaf subproblems that admit no further tripartition $(\overline{\mathcal{G}}_4, \overline{\mathcal{G}}_5, \overline{\mathcal{G}}_6, \overline{\mathcal{G}}_7)$, DICOLA applies a base structure

---

**Algorithm 3:** DICOLA

**Input:** Observed variable set $\mathbf{O}$
**Output:** The PAG $\mathcal{P}$ over $\mathbf{O}$

1   **Function** RECURSIVELEARN($\mathbf{K}$)
2    $(\mathbf{A}, \mathbf{B}, \mathbf{C}, \textit{flag}) \leftarrow$ FINDDECOMPOSITION($\mathbf{K}$);
3    ▷ flag indicates whether the valid decomposition is found
4    **if** $\textit{flag} =$ FALSE **then**
5     $\mathcal{L}_{\mathbf{K}} \leftarrow$ STRUCTURELEARNING($\mathbf{K}$);
6     **return** $\mathcal{L}_{\mathbf{K}}$;
7    $\mathcal{L}_{\mathbf{A} \cup \mathbf{C}} \leftarrow$ RECURSIVELEARN($\mathbf{A} \cup \mathbf{C}$);
8    $\mathcal{L}_{\mathbf{B} \cup \mathbf{C}} \leftarrow$ RECURSIVELEARN($\mathbf{B} \cup \mathbf{C}$);
9    $\mathcal{L}_{\mathbf{K}} \leftarrow$ MERGESKELETONS($\mathcal{L}_{\mathbf{A} \cup \mathbf{C}}, \mathcal{L}_{\mathbf{B} \cup \mathbf{C}}$);
10    **return** $\mathcal{L}_{\mathbf{K}}$;
11   **Function** MERGESKELETONS($\mathcal{L}_{\mathbf{A} \cup \mathbf{C}}, \mathcal{L}_{\mathbf{B} \cup \mathbf{C}}$)
12    $\mathcal{L}_{\mathbf{A} \cup \mathbf{B} \cup \mathbf{C}} \leftarrow \mathcal{L}_{\mathbf{A} \cup \mathbf{C}} \cup \mathcal{L}_{\mathbf{B} \cup \mathbf{C}}$;
13    **foreach** *pair* $(X, Y) \in \mathbf{C}$ **do**
14     **if** $\langle X, Y \rangle \notin \mathcal{L}_{\mathbf{A} \cup \mathbf{C}}$ *or* $\langle X, Y \rangle \notin \mathcal{L}_{\mathbf{B} \cup \mathbf{C}}$ **then**
15      Remove edge $\langle X, Y \rangle$ from $\mathcal{L}_{\mathbf{A} \cup \mathbf{B} \cup \mathbf{C}}$;
16    **return** $\mathcal{L}_{\mathbf{A} \cup \mathbf{B} \cup \mathbf{C}}$;
17 ▷ Main procedure
18   $\mathcal{L}_{\mathbf{O}} \leftarrow$ RECURSIVELEARN($\mathbf{O}$);
19   $\mathcal{P} \leftarrow$ orient $\mathcal{L}_{\mathbf{O}}$ using V-structures and the orientation rules of Zhang (2008b);
20   **return** $\mathcal{P}$

---

learning algorithm (e.g., FCI) on each subset to obtain the corresponding local skeletons $\mathcal{L}_4, \mathcal{L}_5, \mathcal{L}_6, \mathcal{L}_7$.
- **Phase 3: Bottom-up merging.** The local skeletons are merged in a bottom-up manner using MERGESKELETONS (illustrated in Figure 2(e)). Specifically, $\mathcal{L}_4$ and $\mathcal{L}_5$ are merged into $\mathcal{L}_2$, with no edge removal since $|\mathbf{C}| = 1$ for $\overline{\mathcal{G}}_2$. Similarly, $\mathcal{L}_6$ and $\mathcal{L}_7$ are merged to form $\mathcal{L}_3$. Finally, $\mathcal{L}_2$ and $\mathcal{L}_3$ are merged into the global skeleton $\mathcal{L}_1$. Because the separator of $\overline{\mathcal{G}}_1$ is $\mathbf{C} = \{X_4, X_5\}$ and the edge $\langle X_4, X_5 \rangle$ appears in both $\mathcal{L}_2$ and $\mathcal{L}_3$, it is retained in $\mathcal{L}_1$.

Applying the orientation rules for v-structures and the orientation rules in Zhang (2008b) to the resulting skeleton $\mathcal{L}_1$, DICOLA produces the PAG shown in Figure 2(c).

**Theorem 4** (Soundness and Completeness). *Assume access to an oracle for conditional independence tests. If the base structure learning algorithm is sound and complete (e.g., FCI), then* DICOLA *is also sound and complete; that is, it correctly recovers the PAG representing the Markov equivalence class of the underlying MAG.*

**Complexity Analysis.** Let $n = |\mathbf{O}|$ denote the number of observed variables. The complexity of DICOLA consists of two components: (1) the cost of identifying decompositions

via Markov blanket learning, and (2) the cost of learning local skeletons. For the first component, using an MB discovery algorithm such as TC (Pellet & Elisseeff, 2008b), the cost of learning the Markov blanket of one variable among $k$ variables is $\mathcal{O}(k)$. For the second component, if FCI is used as the base learner, the worst-case complexity for a subset of size $k$ is $\mathcal{O}(k^2 2^k)$. Suppose the problem is recursively decomposed into $m$ leaf problems, and let $k_{\max}$ be the size of the largest leaf problem. The total cost of UIG construction across all recursion levels is bounded by $\mathcal{O}(mn^2)$, since each step involves MB learning. Hence, the overall complexity is bounded by $\mathcal{O}(mk_{\max}^2 2^{k_{\max}})$. Because $k_{\max} \ll n$ for decomposable graphs, the exponential term is considerably reduced compared to the global complexity $\mathcal{O}(n^2 2^n)$.

## 6. Experiments

### 6.1. Synthetic Data

In this section, we present empirical results on synthetic data generated from both random graph structures and real-world Bayesian networks drawn from the Bayesian Network Repository[1], a benchmark for causal structure learning.

We consider five causal discovery algorithms that allow latent variables:

- **FCI**: The foundational algorithm for causal discovery in the presence of latent confounders (Spirtes et al., 1995). Implementations are taken from *causal-learn* (Zheng et al., 2024).
- **RFCI**: A computationally more efficient variant of FCI proposed by Colombo et al. (2012). We adopt its from the *pcalg* R package (Kalisch et al., 2012).
- **FCI$^+$**: An optimized variant of FCI specifically designed to reduce the number of CI tests, proposed by Claassen et al. (2013). We adopt its implementation from the *pcalg* R package (Kalisch et al., 2012).
- **ICD**: The iterative causal discovery algorithm introduced by Rohekar et al. (2021). Source code is obtained from the official GitHub repository (https://github.com/IntelLabs/causality-lab).
- **L-MARVEL**: The recursive causal discovery method of Akbari et al. (2021), designed to handle latent confounding. We use the implementation in the Python package *rcd* (Mokhtarian et al., 2025).

For each base algorithm, we compare its standalone performance with its integration within the proposed DI-COLA framework, denoted as DICOLA+Method (e.g., DI-COLA+FCI)[2].

---

[1] https://www.bnlearn.com/bnrepository/

[2] Implementation details and full experimental settings are provided in Appendix C.

**Experimental setup and metrics.** Following the convention in Colombo et al. (2012); Akbari et al. (2021); Rohekar et al. (2021), the underlying causal structures are parameterized as linear Gaussian structural causal models. Causal coefficients are sampled uniformly from $\pm(0.5, 1)$, and all noise terms are independently drawn from a standard Gaussian distribution. We evaluate structural accuracy using Precision, Recall, and F1-score of the recovered edges, following Akbari et al. (2021); Mokhtarian et al. (2025). To assess computational efficiency, we record both the number of conditional independence (CI) tests and the execution time (in seconds) required by each method on each dataset. For all methods, the maximum runtime is capped at ten times the number of observed variables, after which execution is automatically terminated. All reported results are averaged over 50 independently generated datasets to ensure statistical reliability. For each dataset, latent variables are randomly selected from vertices in the underlying causal graph that have at least two children.

**Random structures.** We simulate random underlying causal structures from the Erdős–Rényi model $\mathrm{ER}(n, d)$ (Erdős & Rényi, 1960), where $n$ denotes the number of variables in $\mathbf{V}$ and $d$ specifies the expected average degree of each variable. To evaluate the performance of different methods under varying graph sizes, sparsity levels, and degrees of latent confounding, we consider four settings:

- For graphs with 30 vertices, we use $\mathrm{ER}(30, 3)$ and $\mathrm{ER}(30, 5)$ and randomly designate 3 variables as latent.
- For larger graphs, we use $\mathrm{ER}(50, 3)$ and consider two levels of latent variables, with 5 and 7 latent variables, respectively.

**Benchmark structures.** We further evaluate the compared methods on four benchmark Bayesian networks with varying dimensionality:

- MILDEW: 35 vertices and 3 latent variables;
- BARLEY: 48 vertices and 4 latent variables;
- ANDES: 223 vertices and 10 latent variables;
- LINK: 724 vertices and 50 latent variables.

**Results.** Due to space limitations, we present results for three metrics under one representative random-structure setting and one benchmark structure in Figure 3, with complete results provided in Appendix C. Note that certain methods are excluded from some plots due to excessive runtime. As expected, across all settings, DICOLA significantly reduces the number of CI tests and runtime for all base methods. Regarding structural accuracy, our framework generally improves or matches the F1-scores of the base methods. It is noteworthy that while DICOLA+L-MARVEL shows a slight accuracy trade-off on simpler graphs (e.g., $\mathrm{ER}(50, 3)$ with 5 latent variables; see Figure 3(a)), this trend is re-

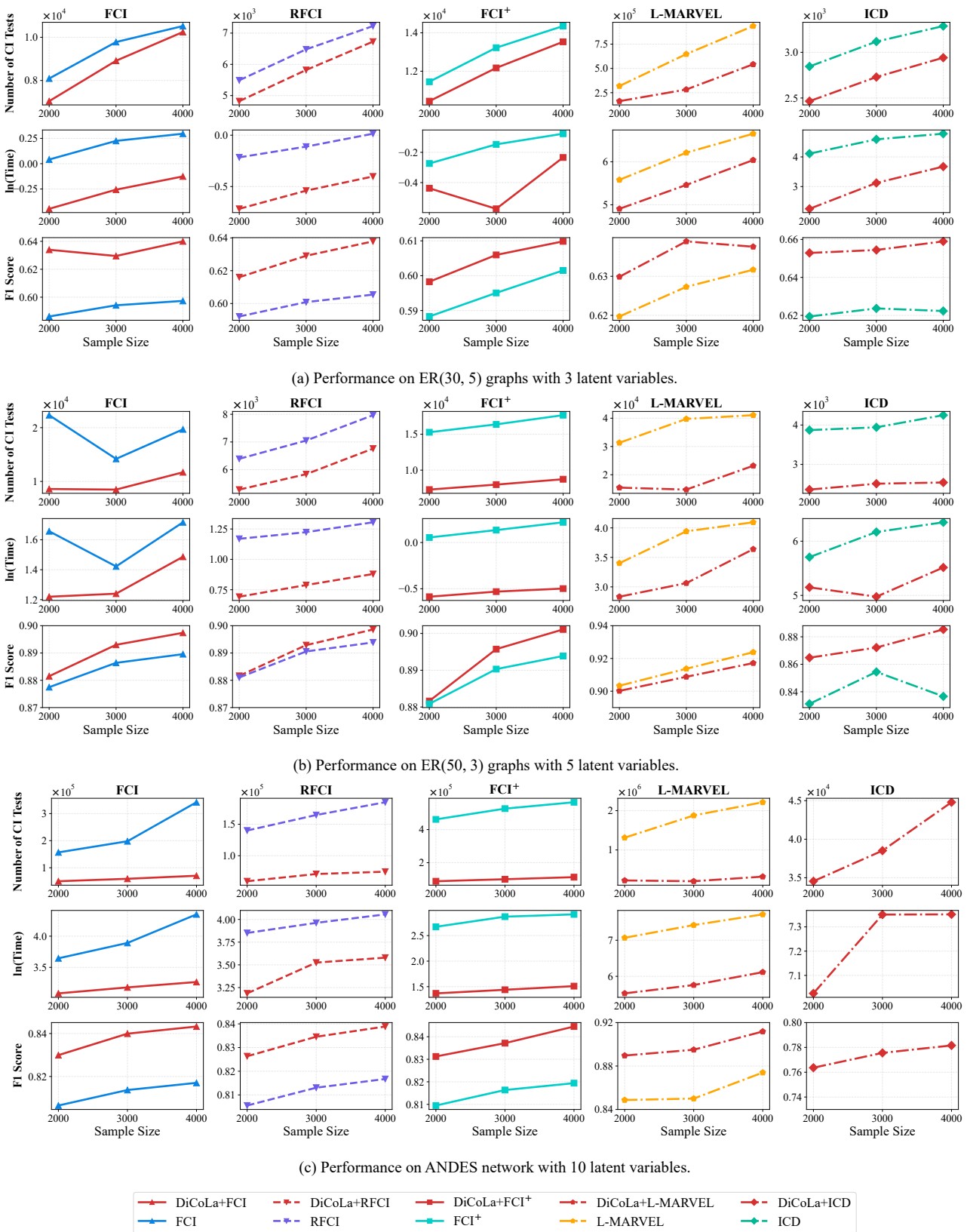

(a) Performance on ER(30, 5) graphs with 3 latent variables.

(b) Performance on ER(50, 3) graphs with 5 latent variables.

(c) Performance on ANDES network with 10 latent variables.

*Figure 3*. Performance comparison on ER(30, 5) graphs, ER(50, 3) graphs, and the real-world ANDES network. Panels (a)–(c) correspond to these three settings, respectively.

versed or mitigated in more challenging settings with higher dimensionality, denser connectivity, or increased latent confounding (e.g., the ANDES network with 10 latent variables; see Figure 3(b)).

### 6.2. Application to *Arabidopsis thaliana* Gene Expression Data

In this section, we apply DICOLA in conjunction with the seminal FCI algorithm (DICOLA+FCI) to the gene expression dataset from Wille et al. (2004). This dataset comprises gene expression measurements from the *Arabidopsis thaliana* collected under 118 different experimental conditions, including variations in light, darkness, and exposure to growth hormones. In *Arabidopsis thaliana*, isoprenoids are synthesized via two distinct pathways located in different cellular compartments: the mevalonate (MVA) pathway in the cytoplasm and the non-mevalonate (MEP) pathway in the chloroplast We focus on 33 genes involved in these pathways or localized in the mitochondrion. While the true causal graph is unknown, biological literature and previous statistical analyses suggest a modular structure where genes within the MEP and MVA pathways are densely connected, with specific loci acting as bridges for cross-talk (Laule et al., 2003; Rodriéguez-Concepcioén et al., 2004; Wille et al., 2004; Frot et al., 2019).

**Results.** Figure 4 shows the adjacency matrix of the PAG recovered by DICOLA+FCI. Comparing our results with the known metabolic pathways and the graphical models reported in Wille et al. (2004, Figure 3), we observe several biologically consistent patterns:

- *(1) Modular Recovery:* DICOLA+FCI successfully recovers the block-diagonal structure characteristic of the two distinct biosynthesis pathways. We observe dense connectivity among genes in the plastidial MEP pathway (e.g., *DXR*, *MCT*, *CMK*, *MECPS*), shown in the upper-left quadrant of the adjacency matrix. Similarly, genes associated with the cytosolic MVA pathway (e.g., *MK*, *MPDC1*, *MPDC2*, *IPPI2*) form a coherent cluster in the lower-right section. This aligns with the findings of Wille et al. (2004), who reported modules of closely connected genes within each pathway.
- *(2) Inter-pathway Connections:* While the pathways operate in distinct compartments, DICOLA+FCI identifies sparse potential interactions between them. These findings corroborate that while intra-pathway regulation involves tight co-expression modules, cross-talk is restricted to specific loci, as previously suggested by Wille et al. (2004); Laule et al. (2003).

## 7. Conclusions and Discussions

In this work, we established the theoretical foundations for divide-and-conquer causal structure learning in the presence

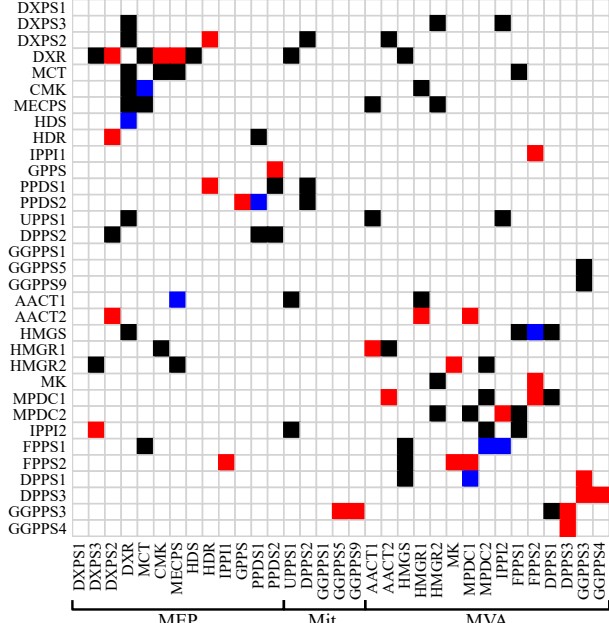

*Figure 4.* Estimated PAG obtained by applying DICOLA+FCI to the *Arabidopsis thaliana* dataset. The adjacency matrix visualizes the edge marks directed from the gene labelled by the $i$-th row to the gene labelled by the $j$-th column: black squares indicate arrowheads, blue squares indicate tails, and red squares indicate circles.

of latent variables by leveraging m-separation properties of maximal ancestral graphs. Building on those theories, we proposed DICOLA, a recursive decomposition framework designed to improve the efficiency of existing causal structure learning methods. We further proved that DICOLA is sound and complete whenever the base structure learning algorithm is sound and complete. Extensive experiments demonstrate that DICOLA substantially improves computational efficiency while preserving structural accuracy.

While DICOLA provides substantial computational gains on sparse and well-decomposable graphs, its acceleration naturally becomes less pronounced as graph density increases. We provide a more detailed discussion of this limitation, together with additional experiments on denser graphs, comparisons with local causal structure learning methods, and relations to methods for learning causal models with latent variables, in Appendix D. Several directions remain for future work. First, developing more effective strategies for identifying valid decompositions, especially in dense graphs where sparse separators may be difficult to find. Second, extending the framework to settings with selection bias. Third, integrating DICOLA with score-based and hybrid causal discovery methods, and establishing theoretical guarantees for the soundness and correctness of the resulting procedures, could further broaden its applicability.

## Acknowledgements

We appreciate the comments from anonymous reviewers, which greatly helped to improve the paper. This research was supported by the National Key R&D Program of China (2022YFA1008300), the National Natural Science Foundation of China (62306019, 62472415), and the GuangDong Basic and Applied Basic Research Foundation (2025A1515010103, 2026B1515020017). Feng Xie was supported by the China Scholarship Council (CSC), the Beijing Key Laboratory of Applied Statistics and Digital Regulation, and the BTBU Digital Business Platform Project by BMEC.

## Impact Statement

This paper presents methodological advancements in the field of Causal Inference, specifically focusing on structure learning in the presence of latent variables. While the broader application of causal discovery has significant potential to impact domains such as healthcare, economics, and policy-making, the contributions of this work are primarily theoretical and algorithmic. We do not foresee any immediate negative societal consequences or specific ethical concerns that require highlighting beyond standard considerations in data science.

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

# Appendix Contents

# A. Detailed Preliminaries

*Table 1.* The list of main symbols used in this paper

| Symbol | Description |
|---|---|
| $\mathcal{D}$ | A directed acyclic graph (DAG) |
| $\mathcal{M}$ | A maximal ancestral graph (MAG) |
| $\mathcal{P}$ | A partial ancestral graph (PAG) |
| $\overline{\mathcal{G}}_{\mathbf{K}}$ | An undirected independence graph of $\mathcal{M}$ relative to $\mathbf{K}$ |
| $\mathcal{L}_{\mathbf{K}}$ | The local skeleton over $\mathbf{K}$ relative to $\mathcal{M}$ |
| $\mathbf{V}$ | The set of all variables |
| $\mathbf{O}$ | The set of observed variables |
| $\mathbf{L}$ | The set of latent variables |
| $|\mathbf{K}|$ | The number of variables in $\mathbf{K}$ |
| $\mathrm{Pa}(X, \mathcal{G})$ | The set of parents of $X$ in the graph $\mathcal{G}$. |
| $\mathrm{MB}_{\mathbf{V}}(X)$ | The Markov blanket of $X$ relative to the variable set $\mathbf{V}$. |
| $\mathrm{An}(X, \mathcal{G})$ | The set of ancestors of $X$ in the graph $\mathcal{G}$. |
| $\mathrm{An}^{+}(X, \mathcal{G})$ | The set of ancestors of $X$, including $X$ itself, in the graph $\mathcal{G}$. |
| $\mathcal{G}[\mathbf{W}]$ | The induced subgraph of $\mathcal{G}$ on $\mathbf{W}$ |

## A.1. Supplementary Terminology

**Graphs.** A graph $\mathcal{G} = (\mathbf{V}, \mathbf{E})$ consists of a set of vertices $\mathbf{V} = \{V_1, \ldots, V_n\}$ and a set of edges $\mathbf{E}$. We consider simple graphs, meaning no self-loops or multiple edges between two vertices. The two ends of an edge are called *marks*, which can be a tail ('−'), an arrowhead ('>'), or a circle ('∘'). For convenience, we use an asterisk ('∗') to denote any allowed edge mark. An edge is *into* (*out of*) $V_i$ if it has an arrowhead (tail) at $V_i$. Given a graph $\mathcal{G} = (\mathbf{V}, \mathbf{E})$ and a subset $\mathbf{W} \subseteq \mathbf{V}$, the *induced subgraph* of $\mathcal{G}$ on $\mathbf{W}$, denoted by $\mathcal{G}[\mathbf{W}]$, is the graph whose vertex set is $\mathbf{W}$ and whose edge set consists of all edges in $\mathcal{G}$ with both endpoints in $\mathbf{W}$. A graph $\mathcal{G}' = (\mathbf{V}', \mathbf{E}')$ is a *subgraph* of $\mathcal{G}$ if $\mathbf{V}' \subseteq \mathbf{V}$ and $\mathbf{E}' \subseteq \mathbf{E}$. For two vertices $X$ and $Y$ in $\mathcal{G}$, $X$ and $Y$ are *adjacent* if there is an edge between them. If $X \to Y / X \leftarrow Y / X \leftrightarrow Y$ in $\mathcal{G}$, then $X$ is a *parent/child/spouse* of $Y$.

**Paths.** A *path* $\pi$ from $X$ to $Y$ in $\mathcal{G}$ is a sequence of distinct variables $\pi = \langle X = V_0, \ldots, Y = V_n \rangle$ such that for $0 \leq i \leq n - 1$, $V_i$ and $V_{i+1}$ are adjacent in $\mathcal{G}$. *The length of a path* equals the number of edges on the path. Given a path $\pi = \langle V_0, V_1, \ldots, V_k \rangle$, the *subpath* of $\pi$ between vertices $V_i$ and $V_j$ ($0 \leq i < j \leq k$) is defined as the sequence of consecutive vertices from $V_i$ to $V_j$ along $\pi$, denoted by $\pi(V_i, V_j)$. For example, if $\pi = \langle V_1, V_2, V_3, V_4, V_5 \rangle$, then $\pi(V_2, V_4) = \langle V_2, V_3, V_4 \rangle$.

**Undirected Graph.** A graph is called an *undirected graph* if all of its edges are undirected (−). In an undirected graph, a vertex $X$ is said to be *separated* from a vertex $Y$ by a set of vertices $\mathbf{Z}$ if every path between $X$ and $Y$ contains at least one vertex in $\mathbf{Z}$. More generally, two vertex sets $\mathbf{X}$ and $\mathbf{Y}$ are said to be separated by $\mathbf{Z}$ if, for every pair $(X, Y)$ with $X \in \mathbf{X}$ and $Y \in \mathbf{Y}$, the vertices $X$ and $Y$ are separated by $\mathbf{Z}$. Given a MAG $\mathcal{M}$ and a vertex set $\mathbf{K}$, the *(local) skeleton* $\mathcal{L}_{\mathbf{K}}$ is the undirected graph over $\mathbf{K}$ that contains an edge between $X$ and $Y$ if and only if no subset $\mathbf{Z} \subseteq \mathbf{K}$ m-separates $X$ and $Y$ in $\mathcal{M}$. When $\mathbf{K} = \mathbf{O}$, the local skeleton coincides with the skeleton of $\mathcal{M}$.

**Set.** A *tripartition* of a set $\mathbf{V}$ is a collection $(\mathbf{V}_1, \mathbf{V}_2, \mathbf{V}_3)$ of three pairwise disjoint subsets whose union is $\mathbf{V}$. If there exists a directed path from $X$ to $Y$, then $X$ is an *ancestor* of $Y$ and $Y$ is a *descendant* of $X$. For a vertex $X$ in a MAG $\mathcal{M}$, let $\mathrm{An}(X, \mathcal{M})$ denote the set of ancestors of $X$ in $\mathcal{M}$, and define $\mathrm{An}^{+}(X, \mathcal{M}) = \mathrm{An}(X, \mathcal{M}) \cup \{X\}$. For a vertex set $\mathbf{X}$, we similarly denote its set of ancestors by $\mathrm{An}(\mathbf{X}, \mathcal{M})$, and define $\mathrm{An}^{+}(\mathbf{X}, \mathcal{M}) = \mathrm{An}(\mathbf{X}, \mathcal{M}) \cup \mathbf{X}$.

**Definition 3** (**Causal Markov Condition** (Spirtes et al., 2000)). *Let $\mathcal{G}$ be a causal graph with vertex set $\mathbf{V}$, and let $P(\mathbf{V})$ denote a probability distribution over $\mathbf{V}$ generated by the causal structure represented by $\mathcal{G}$. We say that $P(\mathbf{V})$ satisfies the* Causal Markov Condition *with respect to $\mathcal{G}$ if, for any triplet of disjoint subsets $\mathbf{X}, \mathbf{Y}, \mathbf{Z} \subseteq \mathbf{V}$, whenever $\mathbf{X}$ and $\mathbf{Y}$ are m-separated by $\mathbf{Z}$ in $\mathcal{G}$, then $\mathbf{X}$ and $\mathbf{Y}$ are conditionally independent given $\mathbf{Z}$ in $P(\mathbf{V})$.*

**Definition 4** (**Causal Faithfulness Condition** (Spirtes et al., 2000)). *Let $\mathcal{G}$ be a causal graph with vertex set $\mathbf{V}$, and let*

$P(\mathbf{V})$ *denote a probability distribution over* $\mathbf{V}$ *generated by the causal structure represented by* $\mathcal{G}$. *We say that* $P(\mathbf{V})$ *satisfies the* Causal Faithfulness Condition *with respect to* $\mathcal{G}$ *if, for any triplet of disjoint subsets* $\mathbf{X}, \mathbf{Y}, \mathbf{Z} \subseteq \mathbf{V}$, *whenever* $\mathbf{X}$ *and* $\mathbf{Y}$ *are conditionally independent given* $\mathbf{Z}$ *in* $P(\mathbf{V})$, *then* $\mathbf{X}$ *and* $\mathbf{Y}$ *are m-separated by* $\mathbf{Z}$ *in* $\mathcal{G}$.

Under these two conditions, the conditional independence relations among the observed variables correspond exactly to m-separation relations in the causal graph. Accordingly, throughout this work we use the notation $\perp\!\!\!\perp$ interchangeably to denote conditional independence in the distribution and m-separation in MAGs.

### A.2. Markov Blanket

The concept of the *Markov blanket* was first coined by Pearl (1988) and has become a widely used technique for reducing the number of variables or features, thereby enabling more efficient and robust model construction (Guyon & Elisseeff, 2003; Pellet & Elisseeff, 2008b; Gao & Ji, 2016). Intuitively, the Markov blanket of a variable $X$ consists of all variables that contain information about $X$ that cannot be obtained from any other variable (Aliferis et al., 2010). Throughout this work, we adopt the convention that the term *Markov blanket* refers to the *minimal* such set[3]. The formal definition of a Markov blanket is given below.

**Definition 5** (**Markov Blanket**). *The Markov blanket of a variable* $X$ *relative to a set of variables* $\mathbf{V}$, *denoted by* $\mathrm{MB}_{\mathbf{V}}(X)$, *is the minimal subset of* $\mathbf{V} \setminus \{X\}$ *such that* $X$ *is conditionally independent of all remaining variables given* $\mathrm{MB}_{\mathbf{V}}(X)$, *formally,*

$$X \perp\!\!\!\perp V \mid \mathrm{MB}_{\mathbf{V}}(X), \quad \forall V \in \mathbf{V} \setminus \big(\mathrm{MB}_{\mathbf{V}}(X) \cup \{X\}\big).$$

Under the causal Faithfulness assumption, the Markov blanket admits a graphical characterization that depends on the underlying causal graph.

**Property 1** (Markov Blanket in DAGs (Pearl, 1988; 2000; 2009)). *Assuming faithfulness, in a directed acyclic graph (DAG), the Markov blanket of a vertex* $X$ *is unique and consists of the parents of* $X$, *the children of* $X$, *and the parents of the children of* $X$.

**Property 2** (Markov Blanket in MAGs (Richardson, 2003; Pellet & Elisseeff, 2008a)). *Assuming faithfulness, in a maximal ancestral graph (MAG), the Markov blanket of a vertex* $X$ *consists of the parents, children, and parents of the children of* $X$, *together with the district of* $X$ *and the districts of the children of* $X$, *as well as the parents of all vertices in these districts. Here, the* district *of a vertex* $V$ *is defined as the set of vertices reachable from* $V$ *using only bidirected edges.*

## B. Proofs

In this section, we provide detailed proofs for the theoretical results presented in the main text. We begin by establishing several fundamental lemmas, followed by the proofs of the main theorems and propositions.

### B.1. Preliminary Lemmas

**Lemma 1.** *Let* $\mathcal{M}$ *be a MAG over* $\mathbf{O}$. *For any three disjoint sets of vertices* $\mathbf{X}, \mathbf{Y}, \mathbf{Z} \subseteq \mathbf{O}$, *and let* $\mathbf{S} = \mathbf{X} \cup \mathbf{Y} \cup \mathbf{Z}$, $(\mathcal{M}[\mathrm{An}^+(\mathbf{S}, \mathcal{M})])^a$ *be the augmented graph of the induced subgraph* $\mathcal{M}[\mathrm{An}^+(\mathbf{S}, \mathcal{M})]$. *Then,* $\mathbf{X}$ *and* $\mathbf{Y}$ *are m-separated by* $\mathbf{Z}$ *in* $\mathcal{M}$ *if and only if* $\mathbf{X}$ *and* $\mathbf{Y}$ *are separated by* $\mathbf{Z}$ *in* $(\mathcal{M}[\mathrm{An}^+(\mathbf{S}, \mathcal{M})])^a$.

*Proof of Lemma 1.* This is a direct consequence of Theorem 3.18 in Richardson & Spirtes (2002). □

**Lemma 2.** *Let* $\mathbf{S}$ *be a subset of vertices in a MAG* $\mathcal{M}$. *For any two vertices* $X$ *and* $Y$ *in* $\mathbf{S}$, $X$ *and* $Y$ *are m-separated by a subset of* $\mathbf{S} \setminus \{X, Y\}$ *in* $\mathcal{M}$ *if and only if they are m-separated by* $\mathrm{An}(\{X, Y\}, \mathcal{M}) \cap \mathbf{S}$.

*Proof of Lemma 2.* Let

$$\mathbf{S}' = \mathrm{An}(\{X, Y\}, \mathcal{M}) \cap \mathbf{S}.$$

Suppose $X$ and $Y$ are not m-separated by $\mathbf{S}'$ in $\mathcal{M}$. Since

$$\mathrm{An}^+(\{X, Y\} \cup \mathbf{S}', \mathcal{M}) = \mathrm{An}^+(\{X, Y\}, \mathcal{M}) \cup \mathrm{An}^+(\mathbf{S}', \mathcal{M}) = \mathrm{An}^+(\{X, Y\}, \mathcal{M}),$$

---

[3]Some authors use the term "Markov blanket" without minimality and reserve "Markov boundary" for the minimal set.

Lemma 1 implies that $X$ and $Y$ are not separated by $\mathbf{S}'$ in $(\mathcal{M}[\text{An}^+(\{X,Y\},\mathcal{M})])^a$. Thus, there exists a path $\pi$ between $X$ and $Y$ in $(\mathcal{M}[\text{An}^+(\{X,Y\},\mathcal{M})])^a$ such that no non-endpoint vertex of $\pi$ lies in $\mathbf{S}'$. Every non-endpoint vertex of $\pi$ is contained in $\text{An}(\{X,Y\},\mathcal{M})$, and $\mathbf{S}' = \text{An}(\{X,Y\},\mathcal{M}) \cap \mathbf{S}$. Hence, no non-endpoint vertex of $\pi$ lies in $\mathbf{S}$. Now consider an arbitrary subset $\mathbf{Z} \subseteq \mathbf{S} \setminus \{X,Y\}$. Because the same path $\pi$ also exists in each augmented graph $(\mathcal{M}[\text{An}^+(\{X,Y\} \cup \mathbf{Z},\mathcal{M})])^a$, the vertices $X$ and $Y$ are not separated by $\mathbf{Z}$ in any such graph. Applying Lemma 1 again, we conclude that $X$ and $Y$ are not m-separated by $\mathbf{Z}$ in $\mathcal{M}$. The other direction is straightforward. This completes the proof. $\qquad\square$

**Lemma 3.** *If $\pi$ is a path m-connecting $X$ and $Y$ given $\mathbf{Z}$ in a MAG $\mathcal{M}$, then every vertex on $\pi$ is in $\text{An}^+(\{X,Y\} \cup \mathbf{Z}, \mathcal{M})$.*

*Proof of Lemma 3.* This is a direct consequence of Lemma 3.13 in Richardson & Spirtes (2002) $\qquad\square$

**Lemma 4.** *Suppose that $\pi$ is a path that connects two non-adjacent vertices $X$ and $Y$ in a MAG $\mathcal{M}$. If $\pi$ is not contained completely in $\text{An}^+(\{X,Y\},\mathcal{M})$, then $\pi$ is m-separated by any subset of $\text{An}(\{X,Y\},\mathcal{M})$.*

*Proof of Lemma 4.* Since $\pi$ is not contained completely in $\text{An}^+(\{X,Y\},\mathcal{M})$ and has length $|\pi| > 1$, there exists a vertex $V$ on $\pi$ such that

$$V \notin \text{An}^+(\{X,Y\},\mathcal{M}).$$

Let $\mathbf{Z} \subseteq \text{An}(\{X,Y\},\mathcal{M})$ be arbitrary. We will show that $\pi$ is not m-connecting $X$ and $Y$ given $\mathbf{Z}$.

First note that in a MAG $\mathcal{M}$, $\text{An}(\cdot,\mathcal{M})$ is closed under taking ancestors, so in particular

$$\text{An}(\text{An}(\{X,Y\},\mathcal{M}),\mathcal{M}) = \text{An}(\{X,Y\},\mathcal{M}).$$

Because $\mathbf{Z} \subseteq \text{An}(\{X,Y\},\mathcal{M})$, monotonicity of $\text{An}(\cdot,\mathcal{M})$ implies

$$\text{An}(\mathbf{Z},\mathcal{M}) \subseteq \text{An}(\text{An}(\{X,Y\},\mathcal{M}),\mathcal{M}) = \text{An}(\{X,Y\},\mathcal{M}).$$

Therefore,

$$\text{An}^+(\{X,Y\} \cup \mathbf{Z}, \mathcal{M}) = \text{An}^+(\{X,Y\},\mathcal{M}).$$

By construction, the vertex $V$ satisfies

$$V \notin \text{An}^+(\{X,Y\},\mathcal{M}) = \text{An}^+(\{X,Y\} \cup \mathbf{Z}, \mathcal{M}).$$

Consequently, by Lemma 3, $\pi$ cannot be m-connecting $X$ and $Y$ given $\mathbf{Z}$. Thus, $\pi$ is m-separated by any subset of $\text{An}(\{X,Y\},\mathcal{M})$. $\qquad\square$

### B.2. Proof of Theorem 1

*Proof of Theorem 1.* Suppose $X$ and $Y$ are m-separated by a subset of $\mathbf{A} \cup \mathbf{B} \cup \mathbf{C}$. By Lemma 2, they are m-separated by

$$\mathbf{Z} = \text{An}(\{X,Y\},\mathcal{M}) \cap (\mathbf{A} \cup \mathbf{B} \cup \mathbf{C}).$$

Define

$$\mathbf{S} = \text{An}(\{X,Y\},\mathcal{M}) \cap (\mathbf{A} \cup \mathbf{C}).$$

We will show that $X$ and $Y$ are m-separated by $\mathbf{S}$. Consider an arbitrary path $\pi$ between $X$ and $Y$.

**Case 1:** If $\pi$ is not contained completely in $\text{An}^+(\{X,Y\},\mathcal{M})$, then by Lemma 4, $\pi$ is m-separated by any subset of $\text{An}(\{X,Y\},\mathcal{M})$, including $\mathbf{S}$.

**Case 2:** Then all vertices on $\pi$ lie in $\text{An}^+(\{X,Y\},\mathcal{M})$. Suppose $\pi$ is not m-separated by $\mathbf{S}$. Since $\mathbf{Z}$ m-separates $\pi$ and $\mathbf{S} \subseteq \mathbf{Z}$, there must be at least one vertex $V \in \{\pi \cap (\mathbf{Z} \setminus \mathbf{S})\} \subseteq \mathbf{B}$. Let $V$ be the closest such vertex to $X$ along $\pi$. Then the subpath $\pi(X,V)$ is also not m-separated by $\mathbf{S}$. Since all non-endpoint vertices on $\pi(X,V)$ are not in $\mathbf{B}$, by $\pi(X,V) \subseteq \pi \subseteq \text{An}(\{X,Y\},\mathcal{M})$ and the definition of $\mathbf{S}$, all non-endpoint vertices on $\pi(X,V)$ belong to $\mathbf{S}$. Thus $\pi(X,V)$ m-connects $X$ and $V$ given $\mathbf{S}$, and therefore also given $\mathbf{C} \cup \mathbf{S}$. However, the condition $\mathbf{A} \perp\!\!\!\perp \mathbf{B} \mid \mathbf{C}$, implies $\{X\} \cup (\mathbf{S} \cap \mathbf{A}) \perp\!\!\!\perp B \mid \mathbf{C}$ for any $B \in \mathbf{B}$. Furthermore, we have $X \perp\!\!\!\perp V \mid \mathbf{C} \cup (\mathbf{S} \cap \mathbf{A})$, since $\mathbf{S} \cap \mathbf{A} \subseteq \mathbf{S}$, then

$X \perp\!\!\!\perp V \mid \mathbf{C} \cup \mathbf{S}$. Which contradicts the existence of the m-connecting subpath $\pi(X, V)$ given $\mathbf{C} \cup \mathbf{S}$. Hence $\pi$ must be m-separated by $\mathbf{S}$.

In conclusion, there is a subset of $\mathbf{A} \cup \mathbf{C}$ that m-separates $X$ and $Y$. The other direction is straightforward. The proof is complete.

$\square$

### B.3. Proof of Theorem 2

**Lemma 5.** *Let $X$ and $Y$ be two non-adjacent vertices in a MAG. Then $X$ and $Y$ are m-separated by a vertex set $\mathbf{Z}$ if and only if every sequence $p = \langle X, \ldots, Y \rangle$ connecting $X$ and $Y$ satisfies at least one of the following conditions:*

*(i) $p$ contains a non-collider that is in $\mathbf{Z}$; or*
*(ii) $p$ contains a collider that is not in $\mathbf{Z}$ and whose descendants are also not in $\mathbf{Z}$.*

*Proof of Lemma 5.* Suppose there exists a sequence connecting $X$ and $Y$ that violates both conditions (i) and (ii). Consider a shortest such sequence. It is straightforward to show that this sequence is a path. By construction, this path is not blocked by $\mathbf{Z}$ and therefore is not m-separated. Hence, $X$ and $Y$ are not m-separated by $\mathbf{Z}$. The converse direction follows directly from the definition of m-separation. $\square$

*Proof of Theorem 2.* If $X$ and $Y$ are m-separated by a subset of $\mathbf{A} \cup \mathbf{C}$ or by a subset of $\mathbf{B} \cup \mathbf{C}$, then they are m-separated by a subset of $\mathbf{A} \cup \mathbf{B} \cup \mathbf{C}$. We prove the other direction.

Suppose $X$ and $Y$ are m-separated by a subset of $\mathbf{A} \cup \mathbf{B} \cup \mathbf{C}$. By Lemma 2, they are m-separated by $\mathbf{Z} = \mathrm{An}(\{X, Y\}, \mathcal{M}) \cap \{\mathbf{A} \cup \mathbf{B} \cup \mathbf{C}\}$. Without loss of generality, assume that $X$ is not an ancestor of $Y$ in $\mathcal{M}$, otherwise we can swap the roles of $X$ and $Y$ in the following proof. Let $\mathbf{S}_1 = \mathrm{An}(\{X, Y\}, \mathcal{M}) \cap \{\mathbf{A} \cup \mathbf{C}\}$ and $\mathbf{S}_2 = \mathrm{An}(\{X, Y\}, \mathcal{M}) \cap \{\mathbf{B} \cup \mathbf{C}\}$. We will show that $X$ and $Y$ are m-separated by either $\mathbf{S}_1$ or $\mathbf{S}_2$. That is, there is not a path $\pi_1$ connecting $X$ and $Y$ in $\{\mathbf{A} \cup \mathbf{C}\}$ not m-separated by $\mathbf{S}_1$ and not a path $\pi_2$ connecting $X$ and $Y$ in $\{\mathbf{B} \cup \mathbf{C}\}$ not m-separated by $\mathbf{S}_2$.

If $\pi_i$ is not contained completely in $\mathrm{An}^+(\{X, Y\}, \mathcal{M})$, then by Lemma 4, $\pi_i$ is m-separated by any subset of $\mathrm{An}(\{X, Y\}, \mathcal{M})$, including $\mathbf{S}_1$ and $\mathbf{S}_2$.

If all vertices on $\pi_i$ lie in $\mathrm{An}^+(\{X, Y\}, \mathcal{M})$, suppose that $\pi_1$ cannot be m-separated by $\mathbf{S}_1$ and $\pi_2$ cannot be m-separated by $\mathbf{S}_2$. Since $\mathbf{Z} = \mathbf{S}_1 \cup \mathbf{S}_2$ can m-separate $\pi_1$ and $\pi_2$, there must be one vertex $V_1 \in \{\pi_1 \cap (\mathbf{Z} \setminus \mathbf{S}_1)\} \subseteq \mathbf{B}$ and one vertex $V_2 \in \{\pi_2 \cap (\mathbf{Z} \setminus \mathbf{S}_2)\} \subseteq \mathbf{A}$. Let $V_1$ and $V_2$ be the closest such vertices to $X$ along $\pi_1$ and $\pi_2$, respectively. Then the subpaths $\pi_1(X, V_1)$ and $\pi_2(X, V_2)$ are also not m-separated by $\mathbf{S}_1$ and $\mathbf{S}_2$, respectively. Connecting these two paths at $X$, forms a sequence $p = \langle V_1, \ldots, X, \ldots, V_2 \rangle$ between $V_1 \in \mathbf{B}$ and $V_2 \in \mathbf{A}$.

For any non-collider vertex on $p$, since $\mathbf{S}_1$ not m-separates $\pi(X, V_1)$, thus this vertex cannot be in $\mathbf{S}_1$ and therefore not in $\mathbf{C}$. Similarly, the non-collider vertices on $\pi(X, V_2)$ cannot be in $\mathbf{C}$. Hence, no non-collider vertex on $p$ is in $\mathbf{C}$. For any collider vertex on $\pi(X, V_1)/\pi(X, V_2)$, since $\mathbf{S}_1/\mathbf{S}_2$ not m-separates $\pi(X, V_1)/\pi(X, V_2)$, thus this vertex or its descendant must be in $\mathbf{S}_1/\mathbf{S}_2$. Since these vertices are in $\mathrm{An}(\{X, Y\}, \mathcal{M})$, then every such vertex's descendants must include $X$ or $Y$, which are in $\mathbf{C}$. Hence, all collider vertcies on $\pi(X, V_1)/\pi(X, V_2)$ itself or its descendants are in $\mathbf{C}$.

Now we show that vertex $X$ is also a collider on $p$. Let $Q_1$ be the vertex adjacent to $X$ on $\pi_1(X, V_1)$ and $Q_2$ be the vertex adjacent to $X$ on $\pi_2(X, V_2)$. And $Q_1/Q_2$ must be an ancestor of $X$ or $Y$, since $\pi_1(X, V_1)/\pi_2(X, V_2)$ is contained completely in $\mathrm{An}^+(\{X, Y\}, \mathcal{M})$. If the edge between $X$ and $Q_1$ is out of $X$, by assuming no selection bias, it will be $X \to Q_1$, then $Q_1$ must be an ancestor of $Y$. Which contradicts the assumption that $X$ is not an ancestor of $Y$. Thus, the edge between $X$ and $Q_1$ must be into $X$. Similarly, the edge between $X$ and $Q_2$ must be into $X$. Therefore, $X$ is a collider on $p$. And since $X \in \mathbf{C}$, then $X$ or its descendants are in $\mathbf{C}$. Hence, all collider vertcies on $p$ itself or its descendants are in $\mathbf{C}$ but no non-collider vertex on $p$ is in $\mathbf{C}$. By Lemma 5, $V_1$ and $V_2$ are not m-separated by $\mathbf{C}$, which contradicts the condition $\mathbf{A} \perp\!\!\!\perp \mathbf{B} \mid \mathbf{C}$. Thus, either $\pi_1$ is m-separated by $\mathbf{S}_1$ or $\pi_2$ is m-separated by $\mathbf{S}_2$. This completes the proof.

$\square$

## B.4. Proof of Theorem 3

*Proof of Theorem 3.* Suppose $(X, Y)$ is non-adjacent in $\mathcal{L}_\mathbf{K}$. Then by definition, $X$ and $Y$ are m-separated by a subset of $\mathbf{K}$. If $X \in \mathbf{A}/X \in \mathbf{B}$, and $Y \in \mathbf{A} \cup \mathbf{C}/Y \in \mathbf{B} \cup \mathbf{C}$, by Theorem 1, $X$ and $Y$ are m-separated by a subset of $\mathbf{A} \cup \mathbf{C}/\mathbf{B} \cup \mathbf{C}$. If $X, Y \in \mathbf{C}$, by Theorem 2, $X$ and $Y$ are m-separated by a subset of $\mathbf{A} \cup \mathbf{C}$ or by a subset of $\mathbf{B} \cup \mathbf{C}$. Therefore, $(X, Y)$ is non-adjacent in either $\mathcal{L}_{\mathbf{A} \cup \mathbf{C}}$ or $\mathcal{L}_{\mathbf{B} \cup \mathbf{C}}$. Suppose that $(X, Y)$ are adjacent in $\mathcal{L}_\mathbf{K}$. Then, by Theorems 1 and 2, they are also adjacent in either $\mathcal{L}_{\mathbf{A} \cup \mathbf{C}}$ or $\mathcal{L}_{\mathbf{B} \cup \mathbf{C}}$, or in both. This completes the proof. $\square$

## B.5. Proof of Propositions 1 to 3

**Lemma 6.** *Let $\overline{\mathcal{G}} = (\mathbf{V}, \mathbf{E})$ be an undirected graph with vertex set $\mathbf{V}$ and edge set $\mathbf{E}$. Let $\overline{\mathcal{G}'} = (\mathbf{S}, \mathbf{E}')$ be a subgraph of $\overline{\mathcal{G}}$, where $\mathbf{S} \subseteq \mathbf{V}$, $\mathbf{E}' \subseteq \mathbf{E}$, and every edge in $\mathbf{E}'$ has both endpoints in $\mathbf{S}$. For any three pairwise disjoint vertex sets $\mathbf{X}, \mathbf{Y}, \mathbf{Z} \subseteq \mathbf{S}$, if $\mathbf{X}$ and $\mathbf{Y}$ are separated by $\mathbf{Z}$ in $\overline{\mathcal{G}}$, then $\mathbf{X}$ and $\mathbf{Y}$ are also separated by $\mathbf{Z}$ in $\overline{\mathcal{G}'}$.*

*Proof of Lemma 6.* Suppose that $\mathbf{X}$ and $\mathbf{Y}$ are not separated by $\mathbf{Z}$ in the subgraph $\overline{\mathcal{G}'}$. Then there exists a path $\pi$ in $\overline{\mathcal{G}'}$ connecting a vertex in $\mathbf{X}$ to a vertex in $\mathbf{Y}$ such that no vertex on $\pi$ belongs to $\mathbf{Z}$. Since $\overline{\mathcal{G}'}$ is a subgraph of $\overline{\mathcal{G}}$, the same path $\pi$ also exists in $\overline{\mathcal{G}}$. This contradicts the assumption that $\mathbf{X}$ and $\mathbf{Y}$ are separated by $\mathbf{Z}$ in $\overline{\mathcal{G}}$. Therefore, $\mathbf{X}$ and $\mathbf{Y}$ must be separated by $\mathbf{Z}$ in the subgraph $\overline{\mathcal{G}'}$. $\square$

**Lemma 7.** *Let $\mathcal{M}$ be a MAG over $\mathbf{O}$, and let $\mathbf{S} \subseteq \mathbf{O}$ be a subset of vertices. Denote by $(\mathcal{M}[\mathbf{S}])^a$ the augmented graph of the induced subgraph $\mathcal{M}[\mathbf{S}]$, and by $(\mathcal{M})^a[\mathbf{S}]$ the induced subgraph of the global augmented graph $(\mathcal{M})^a$ restricted to $\mathbf{S}$. Then,*

$$(\mathcal{M}[\mathbf{S}])^a \subseteq (\mathcal{M})^a[\mathbf{S}].$$

*That is, every edge in $(\mathcal{M}[\mathbf{S}])^a$ is also present in $(\mathcal{M})^a[\mathbf{S}]$.*

*Proof.* Suppose that there exists an edge $\langle X, Y \rangle$ in $(\mathcal{M}[\mathbf{S}])^a$. By the definition of the augmented graph, in the induced subgraph $\mathcal{M}[\mathbf{S}]$, the vertices $X$ and $Y$ are either (i) adjacent, or (ii) connected by a collider path $\pi = \langle X = V_0, V_1, \ldots, V_n = Y \rangle$, where $V_i \in \mathbf{S}$ for all $0 \leq i \leq n$.

- If $X$ and $Y$ are adjacent in $\mathcal{M}[\mathbf{S}]$, then they are also adjacent in the global graph $\mathcal{M}$.

- If $X$ and $Y$ are connected by a collider path $\pi$ in $\mathcal{M}[\mathbf{S}]$, then the same collider path $\pi$ also exists in the global graph $\mathcal{M}$.

In both cases, $X$ and $Y$ are collider-connected in $\mathcal{M}$, which implies that the edge $\langle X, Y \rangle$ is present in the global augmented graph $(\mathcal{M})^a$. Since $X, Y \in \mathbf{S}$, this edge is preserved in the induced subgraph $(\mathcal{M})^a[\mathbf{S}]$. Hence, $(\mathcal{M}[\mathbf{S}])^a \subseteq (\mathcal{M})^a[\mathbf{S}]$. $\square$

**Remark 3.** *It is important to note that the converse of Lemma 7 does not generally hold, i.e.,*

$$(\mathcal{M})^a[\mathbf{S}] \not\subseteq (\mathcal{M}[\mathbf{S}])^a.$$

*To see this, consider the simple MAG $\mathcal{M} = X \to Z \leftarrow Y$ and let $\mathbf{S} = \{X, Y\}$. The global augmented graph $(\mathcal{M})^a$ contains the undirected edges $X - Z - Y$ as well as $X - Y$, and hence $(\mathcal{M})^a[\mathbf{S}]$ consists of the single edge $X - Y$. In contrast, the induced subgraph $\mathcal{M}[\mathbf{S}]$ contains no edges, and therefore its augmented graph $(\mathcal{M}[\mathbf{S}])^a$ also has no edges.*

*Thus, the edge $X - Y$ appears in $(\mathcal{M})^a[\mathbf{S}]$ but not in $(\mathcal{M}[\mathbf{S}])^a$, implying that*

$$(\mathcal{M})^a[\mathbf{S}] \neq (\mathcal{M}[\mathbf{S}])^a.$$

*Nevertheless, this discrepancy does not affect our subsequent results.*

*Proof of Proposition 1.* We first show that $(\mathcal{M})^a$ is an undirected independence graph of $\mathcal{M}$. Let $\mathbf{X}, \mathbf{Y}, \mathbf{Z} \subseteq \mathbf{O}$ be three pairwise disjoint vertex sets such that $\mathbf{X}$ and $\mathbf{Y}$ are separated by $\mathbf{Z}$ in $(\mathcal{M})^a$. Define $\mathbf{S} = \mathbf{X} \cup \mathbf{Y} \cup \mathbf{Z}$. By Lemma 7, the augmented graph $(\mathcal{M}[\mathrm{An}^+(\mathbf{S}, \mathcal{M})])^a$ is a subgraph of $(\mathcal{M})^a$. Applying Lemma 6, it follows that $\mathbf{X}$ and $\mathbf{Y}$ are also separated by $\mathbf{Z}$ in $(\mathcal{M}[\mathrm{An}^+(\mathbf{S}, \mathcal{M})])^a$. Then, by Lemma 1, $\mathbf{X}$ and $\mathbf{Y}$ are m-separated by $\mathbf{Z}$ in $\mathcal{M}$. Therefore, $(\mathcal{M})^a$ is an undirected independence graph of $\mathcal{M}$.

Next, we establish the minimality of $(\mathcal{M})^a$. Suppose that removing an edge from $(\mathcal{M})^a$ yields a graph that is still an undirected independence graph of $\mathcal{M}$. Let $\langle X, Y \rangle$ denote the removed edge, and let $\mathbf{Z} = \mathbf{O} \setminus \{X, Y\}$. Since $X$ and $Y$ are collider-connected in $\mathcal{M}$, they are not m-separated given $\mathbf{Z}$. However, after removing the edge $\langle X, Y \rangle$, the vertices $X$ and $Y$ become separated by $\mathbf{Z}$ in the resulting undirected graph, which contradicts the assumption that the graph remains an undirected independence graph of $\mathcal{M}$. Hence, $(\mathcal{M})^a$ is the minimal undirected independence graph of $\mathcal{M}$. $\qquad \square$

*Proof of Proposition 2.* By definition of the augmented graph, $Y$ is adjacent to $X$ in $(\mathcal{M})^a$ if and only if $X$ and $Y$ are collider-connected in $\mathcal{M}$. By Property 2, collider-connectedness between $X$ and $Y$ is equivalent to $Y$ belonging to the Markov blanket of $X$ in $\mathcal{M}$. The claim therefore follows. $\qquad \square$

**Lemma 8.** *(Mokhtarian et al., 2025; Richardson & Spirtes, 2002) Let $\mathcal{M}$ be a MAG over $\mathbf{O}$, and let $\mathbf{S} \subseteq \mathbf{O}$, and $\mathcal{M}_\mathbf{S}$ be the marginal MAG over $\mathbf{S}$. For any three disjoint sets of vertices $\mathbf{X}, \mathbf{Y}, \mathbf{Z} \subseteq \mathbf{S}$, $\mathbf{X}$ and $\mathbf{Y}$ are m-separated by $\mathbf{Z}$ in $\mathcal{M}_\mathbf{S}$ if and only if $\mathbf{X}$ and $\mathbf{Y}$ are m-separated by $\mathbf{Z}$ in $\mathcal{M}$.*

*Proof of Proposition 3.* By construction, the output $\overline{\mathcal{G}}_\mathbf{K}$ of Algorithm 1 is derived from the Markov blankets of variables within $\mathbf{K}$. Thus, we have $\overline{\mathcal{G}}_\mathbf{K} = (\mathcal{M}_\mathbf{K})^a$ by Proposition 2. We first prove that $(\mathcal{M}_\mathbf{K})^a$ is an undirected independence graph of $\mathcal{M}$ relative to $\mathbf{K}$. For any three disjoint sets of vertices $\mathbf{X}, \mathbf{Y}, \mathbf{Z} \subseteq \mathbf{K}$, if $\mathbf{X}$ and $\mathbf{Y}$ are separated by $\mathbf{Z}$ in $(\mathcal{M}_\mathbf{K})^a$, then $\mathbf{X}$ and $\mathbf{Y}$ are m-separated by $\mathbf{Z}$ in $\mathcal{M}_\mathbf{K}$ by Proposition 1. By Lemma 8, $\mathbf{X}$ and $\mathbf{Y}$ are also m-separated by $\mathbf{Z}$ in $\mathcal{M}$. Thus, $(\mathcal{M}_\mathbf{K})^a$ is an undirected independence graph of $\mathcal{M}$ relative to $\mathbf{K}$.

Then, we prove that $(\mathcal{M}_\mathbf{K})^a$ is a minimal undirected independence graph of $\mathcal{M}$ relative to $\mathbf{K}$. Assume the resulting graph that removed an edge $\langle X, Y \rangle$ in $(\mathcal{M}_\mathbf{K})^a$ is still an undirected independence graph of $\mathcal{M}$ relative to $\mathbf{K}$. Let $\mathbf{Z} = \mathbf{K} \setminus \{X, Y\}$. In the resulting graph, $X$ and $Y$ are separated by $\mathbf{Z}$. However, in $\mathcal{M}_\mathbf{K}$, $X$ and $Y$ are not m-separated by $\mathbf{Z}$ since $X$ is in the Markov blanket of $Y$ in $\mathcal{M}_\mathbf{K}$ by Proposition 2, i.e., they are collider connected in $\mathcal{M}_\mathbf{K}$. By Lemma 8, $X$ and $Y$ are also not m-separated by $\mathbf{Z}$ in $\mathcal{M}$. This contradicts the assumption that the resulting graph is still an undirected independence graph of $\mathcal{M}$ relative to $\mathbf{K}$. Therefore, $(\mathcal{M}_\mathbf{K})^a$ is a minimal undirected independence graph of $\mathcal{M}$ relative to $\mathbf{K}$. $\qquad \square$

## B.6. Proof of Theorem 4

*Proof of Theorem 4.* The proof proceeds in two steps: (1) we show that DICOLA recovers the correct global skeleton over $\mathbf{O}$, and (2) we show that the subsequent orientation phase yields the correct PAG representing the Markov equivalence class of the underlying MAG.

***Soundness and Completeness of the Skeleton.*** We establish the soundness and completeness of the skeleton by analyzing each stage of DICOLA.

*Top-down decomposition.* By Proposition 3, all tripartitions $(\mathbf{A}, \mathbf{B}, \mathbf{C})$ returned by FINDDECOMPOSITION across the decomposition tree satisfy $\mathbf{A} \perp\!\!\!\perp \mathbf{B} \mid \mathbf{C}$ in the underlying MAG. Therefore, each recursive decomposition performed by DICOLA is sound.

*Local skeleton learning.* Consider a leaf problem of the decomposition tree corresponding to a subset $\mathbf{K} \subseteq \mathbf{O}$. By construction, no valid tripartition exists for $\mathbf{K}$, and DICOLA directly invokes the base structure learning algorithm (e.g., FCI) on $\mathbf{K}$. Since the base algorithm is assumed to be sound and complete, it correctly recovers the local skeleton over $\mathbf{K}$. That is, an undirected edge between $X, Y \in \mathbf{K}$ is present in the local skeleton if and only if there exists no subset $\mathbf{Z} \subseteq \mathbf{K}$ such that $X \perp\!\!\!\perp Y \mid \mathbf{Z}$.

*Bottom-up merging.* Since the local skeletons at the leaf problems are correct, it follows from Theorem 3 that the merged skeleton at each internal problem is also correct. In particular, for a parent problem with variable set $\mathbf{A} \cup \mathbf{B} \cup \mathbf{C}$, the merged skeleton $\mathcal{L}_{\mathbf{A} \cup \mathbf{B} \cup \mathbf{C}}$ contains an undirected edge between $X$ and $Y$ if and only if there exists no subset $\mathbf{Z} \subseteq \mathbf{A} \cup \mathbf{B} \cup \mathbf{C}$ such that $X \perp\!\!\!\perp Y \mid \mathbf{Z}$.

By merging the local skeletons at the leaf problems, the final merged skeleton at the root $\mathbf{O}$ coincides with the true skeleton of the underlying MAG.

***Soundness and Completeness of Orientation.*** After the skeleton recovery phase, DICOLA has obtained the correct global skeleton $\mathcal{L}_\mathbf{O}$. Moreover, by Theorems 1 to 3, all non-adjacent vertex pairs identified during local skeleton learning are associated with correct separating sets. In addition, by Proposition 3, all non-adjacent vertex pairs identified during the decomposition phase also have correct separating sets.

Given that (i) the global skeleton $\mathcal{L}_\mathbf{O}$ is correct, and (ii) every non-adjacent pair of vertices in $\mathcal{L}_\mathbf{O}$ is associated with a correct separating set, the standard orientation rules for PAGs (Zhang, 2008b) are guaranteed to correctly identify all invariant edge marks. Consequently, DICOLA outputs the PAG corresponding to the Markov equivalence class of the underlying MAG. □

## C. Supplementary Experiments

All experiments are conducted on a machine equipped with an Intel Core i9–14900K CPU and 64 GB of RAM. For Markov Blanket discovery, we employ the Total Conditioning (TC) algorithm (Pellet & Elisseeff, 2008b) as implemented in the *rcd* package (Mokhtarian et al., 2025) [4]. Subsequently, to identify vertex separators within the UIG, we utilize the junction tree routines provided by the *NetworkX* library (Hagberg et al., 2008). For conditional independence testing, we apply the Fisher Z-transformation (Fisher, 1915) with a significance level of $\alpha = 0.01$ across all methods. Our source code is included in the supplementary materials.

All experimental results are summarized in Table 2, with corresponding figures presented below. Marginalizing latent variables may substantially densify the induced MAGs relative to the underlying generating DAGs (Richardson & Spirtes, 2002; Zhang, 2008b); therefore, we also report the average degrees of the induced MAGs in Table 2. More details of those real-world structures can be found at https://www.bnlearn.com/bnrepository/.

*Table 2.* Summary of experimental settings and corresponding results.

| Random Structures | | | | | |
|---|---|---|---|---|---|
| **Underlying Structure** | **\|V\|** | **\|L\|** | **Avg. Degree in DAGs** | **Avg. Degree in Induced MAGs** | **Corresponding Results** |
| ER(30, 3) graphs | 30 | 3 | 3.00 | 3.81 | Figure 5 |
| ER(30, 5) graphs | 30 | 3 | 5.00 | 7.28 | Figure 6 |
| ER(50, 3) graphs | 50 | 5 | 3.00 | 3.83 | Figure 7 |
| ER(50, 3) graphs | 50 | 7 | 3.00 | 4.31 | Figure 8 |

| Real-World Structures | | | | | |
|---|---|---|---|---|---|
| **Underlying Structure** | **\|V\|** | **\|L\|** | **Avg. Degree in DAGs** | **Avg. Degree in Induced MAGs** | **Corresponding Results** |
| MILDEW network | 36 | 3 | 2.63 | 2.73 | Figure 9 |
| BARLEY network | 48 | 4 | 3.50 | 4.03 | Figure 10 |
| ANDES network | 223 | 10 | 3.03 | 3.53 | Figure 11 |
| LINK network | 724 | 50 | 3.11 | 3.21 | Figure 12 |

---

[4]The framework can also be combined with more recent MB discovery methods designed for high-dimensional settings, such as STMB (Gao & Ji, 2016), EAMB (Guo et al., 2022), and CFS-MI (Ling et al., 2022), as well as methods that use smaller conditioning sets, such as MMMB (Tsamardinos et al., 2003) and HITON-MB (Aliferis et al., 2003).

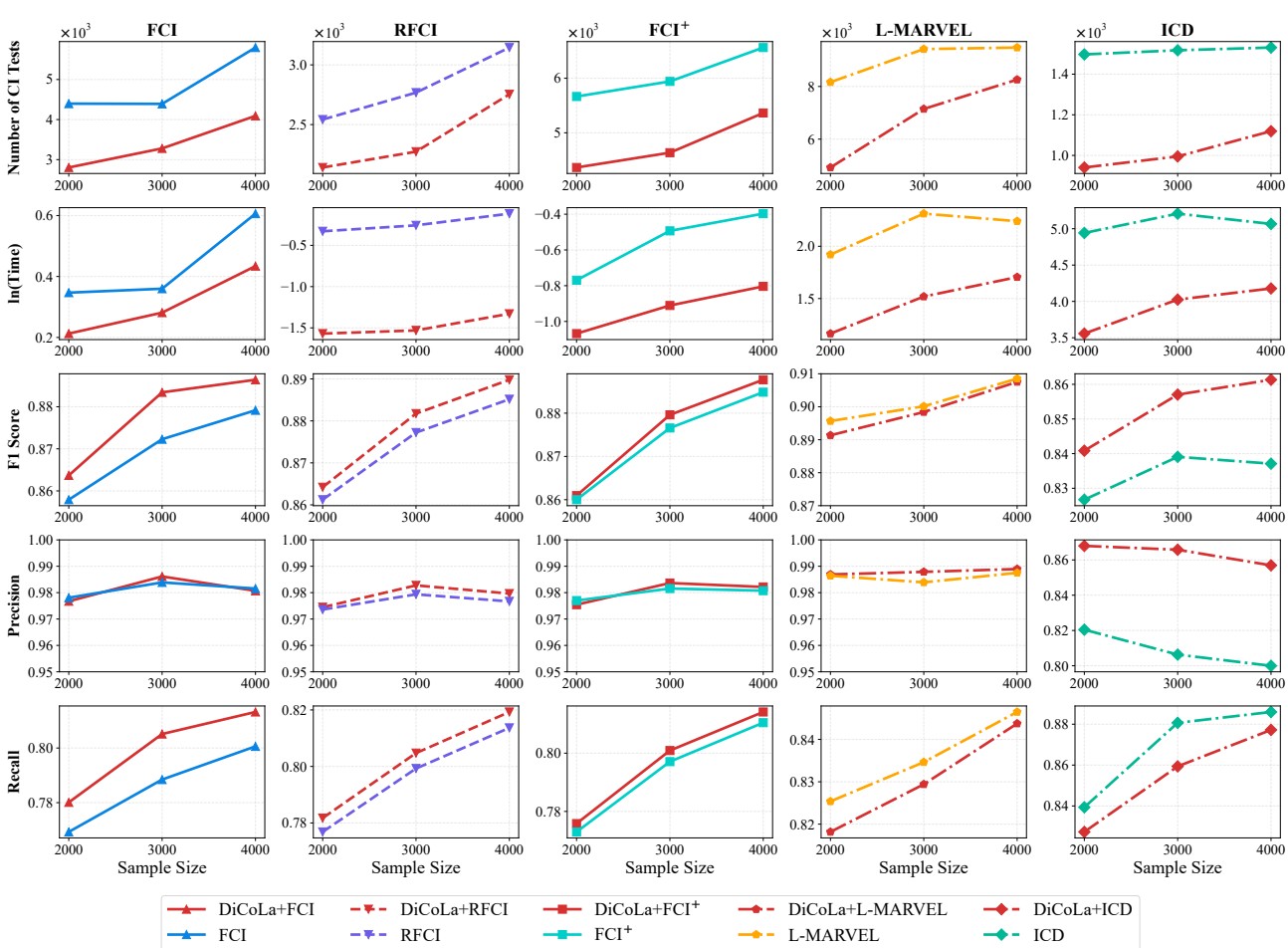

*Figure 5.* Performance on ER(30, 3) graphs with 3 latent variables.

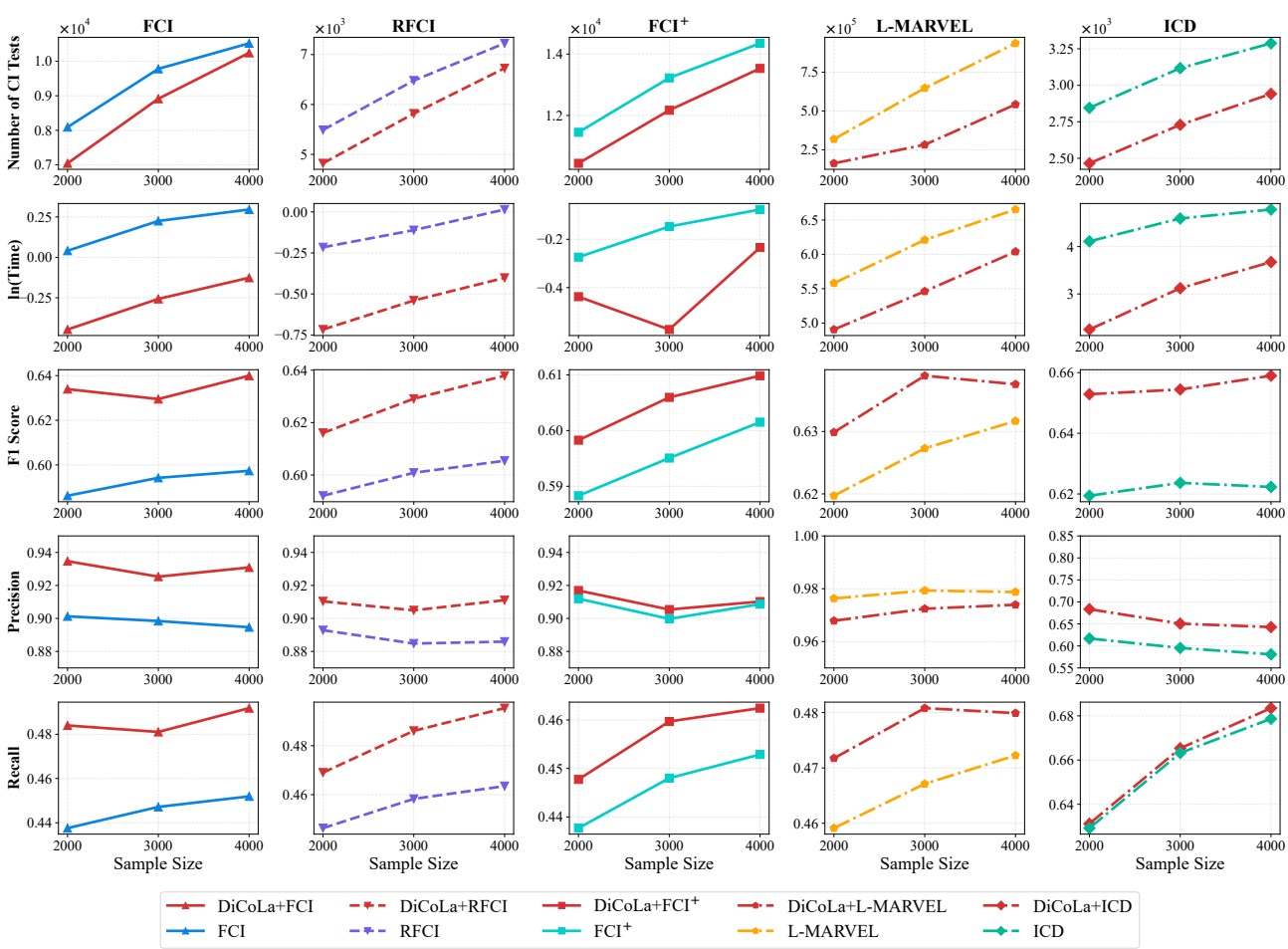

*Figure 6.* Performance on ER(30, 5) graphs with 3 latent variables.

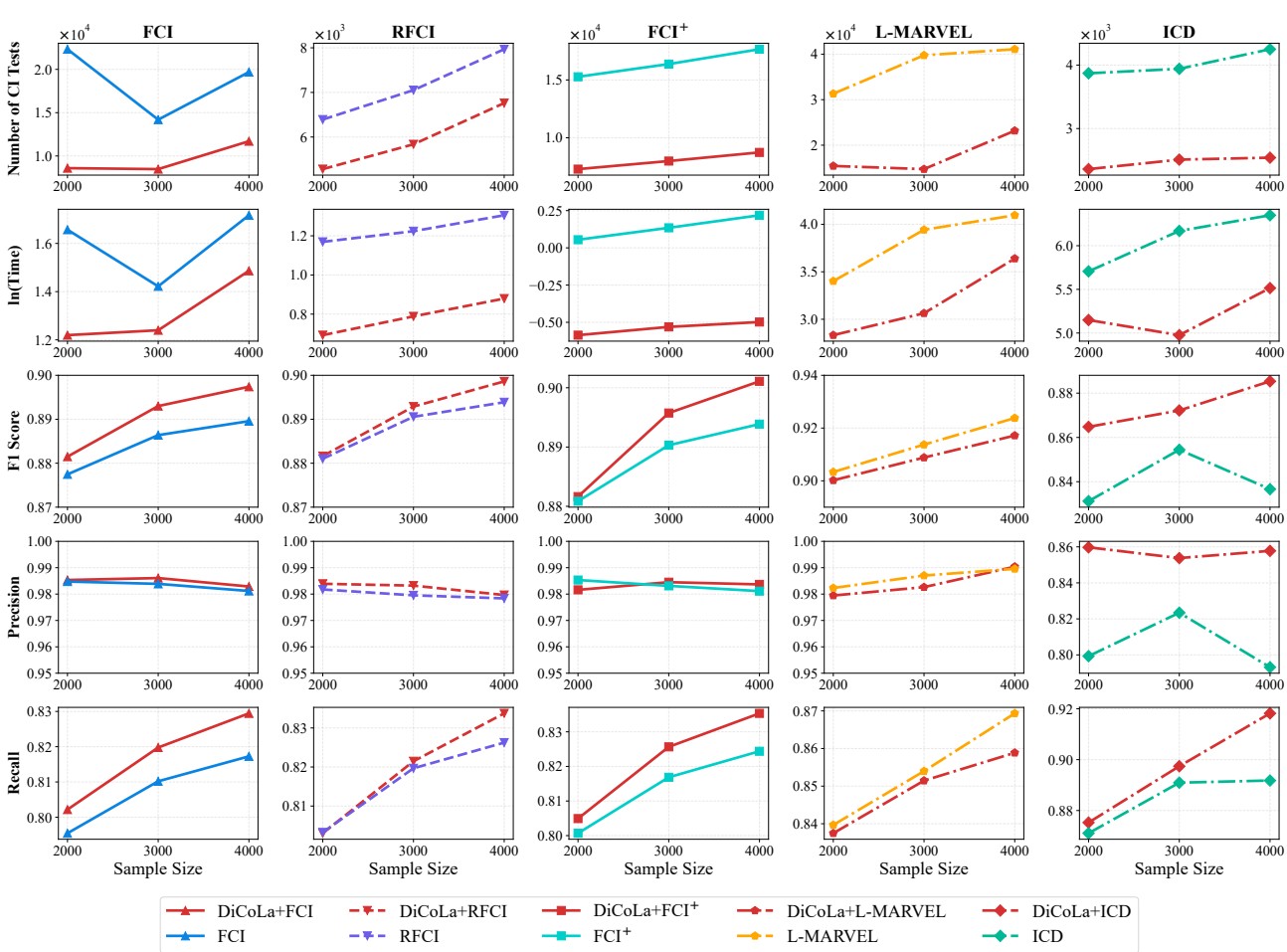

*Figure 7.* Performance on ER(50, 3) graphs with 5 latent variables.

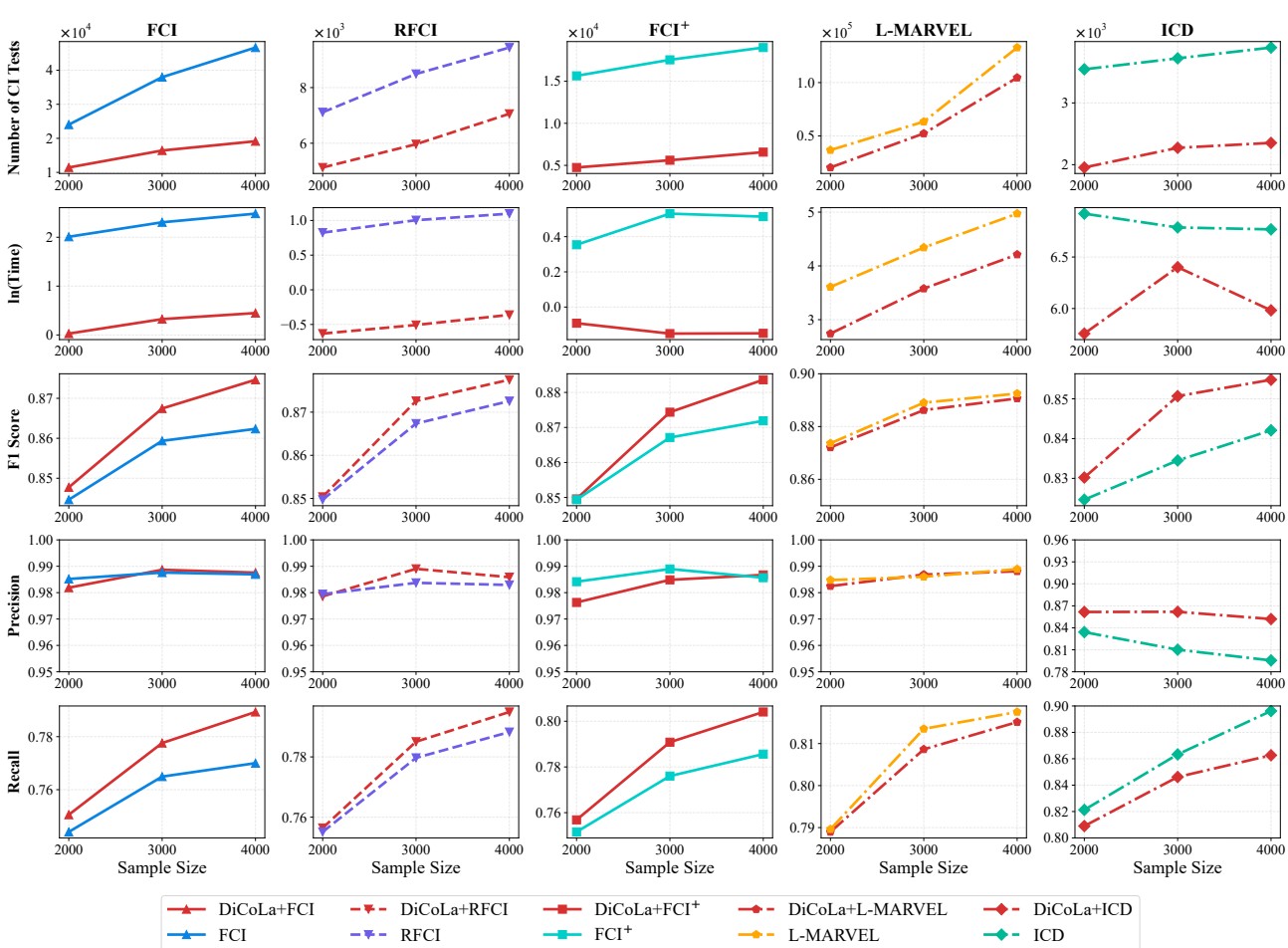

*Figure 8.* Performance on ER(50, 3) graphs with 7 latent variables.

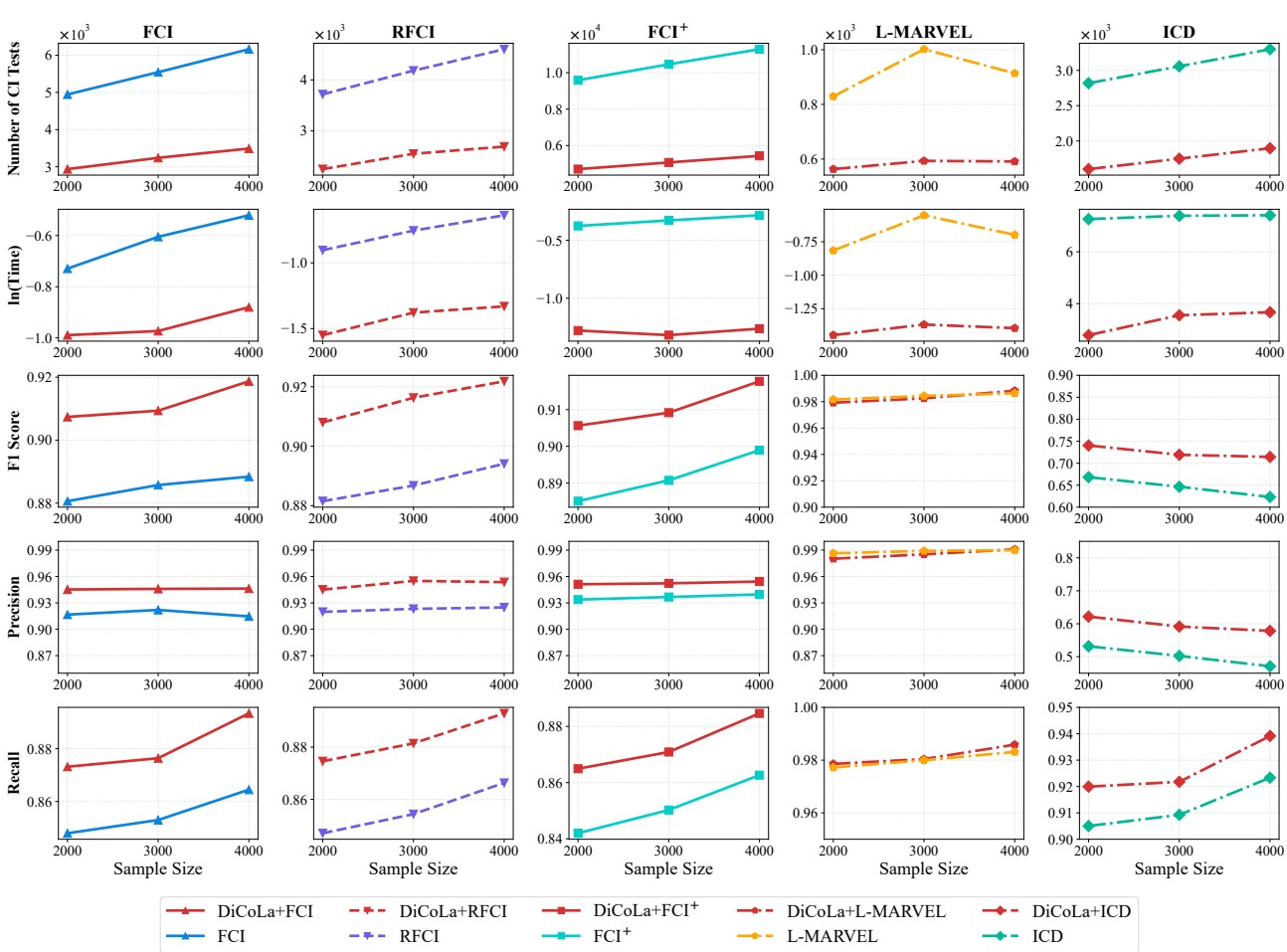

*Figure 9.* Performance on MILDEW network with 3 latent variables.

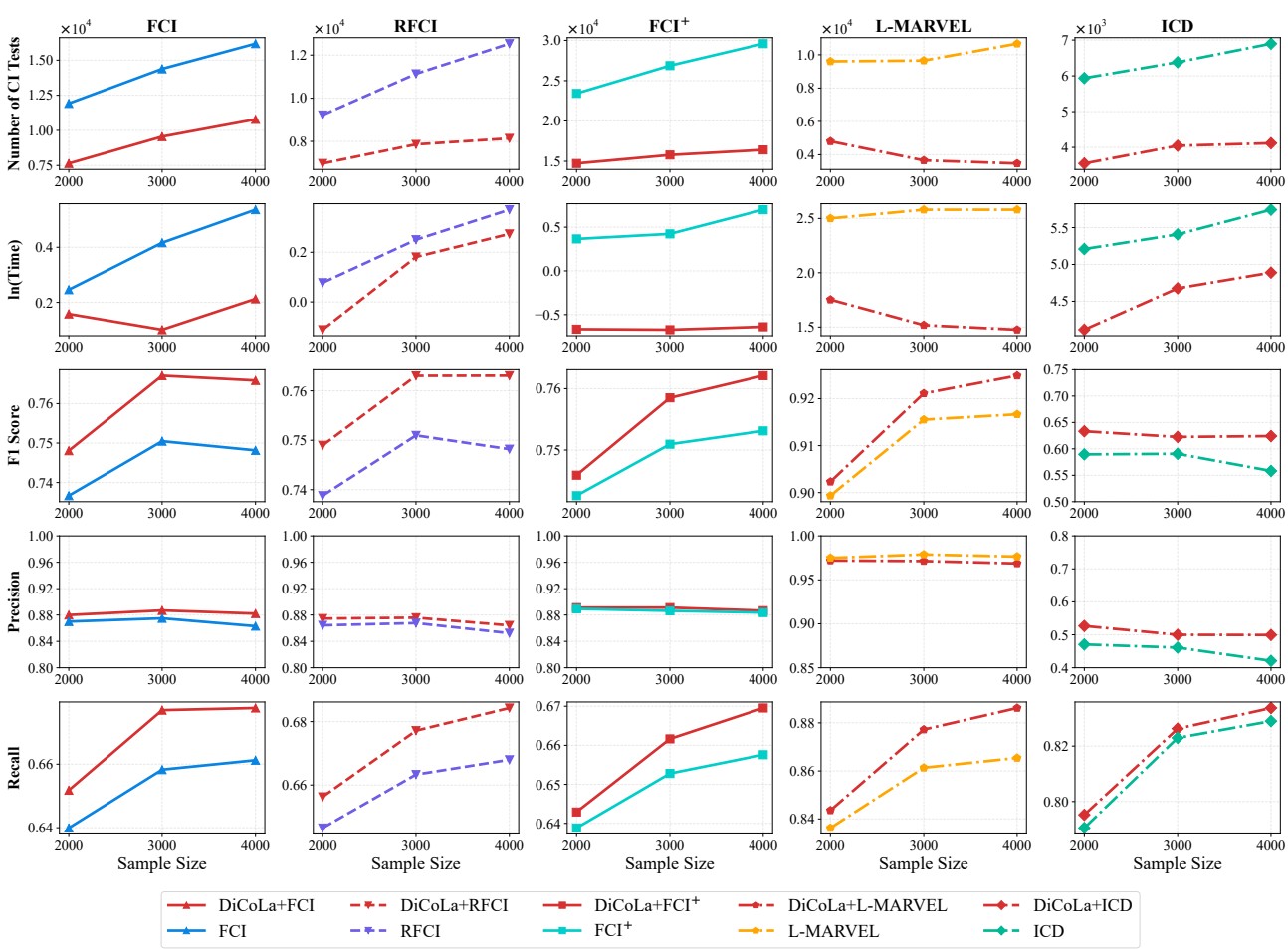

*Figure 10.* Performance on BARLEY network with 4 latent variables.

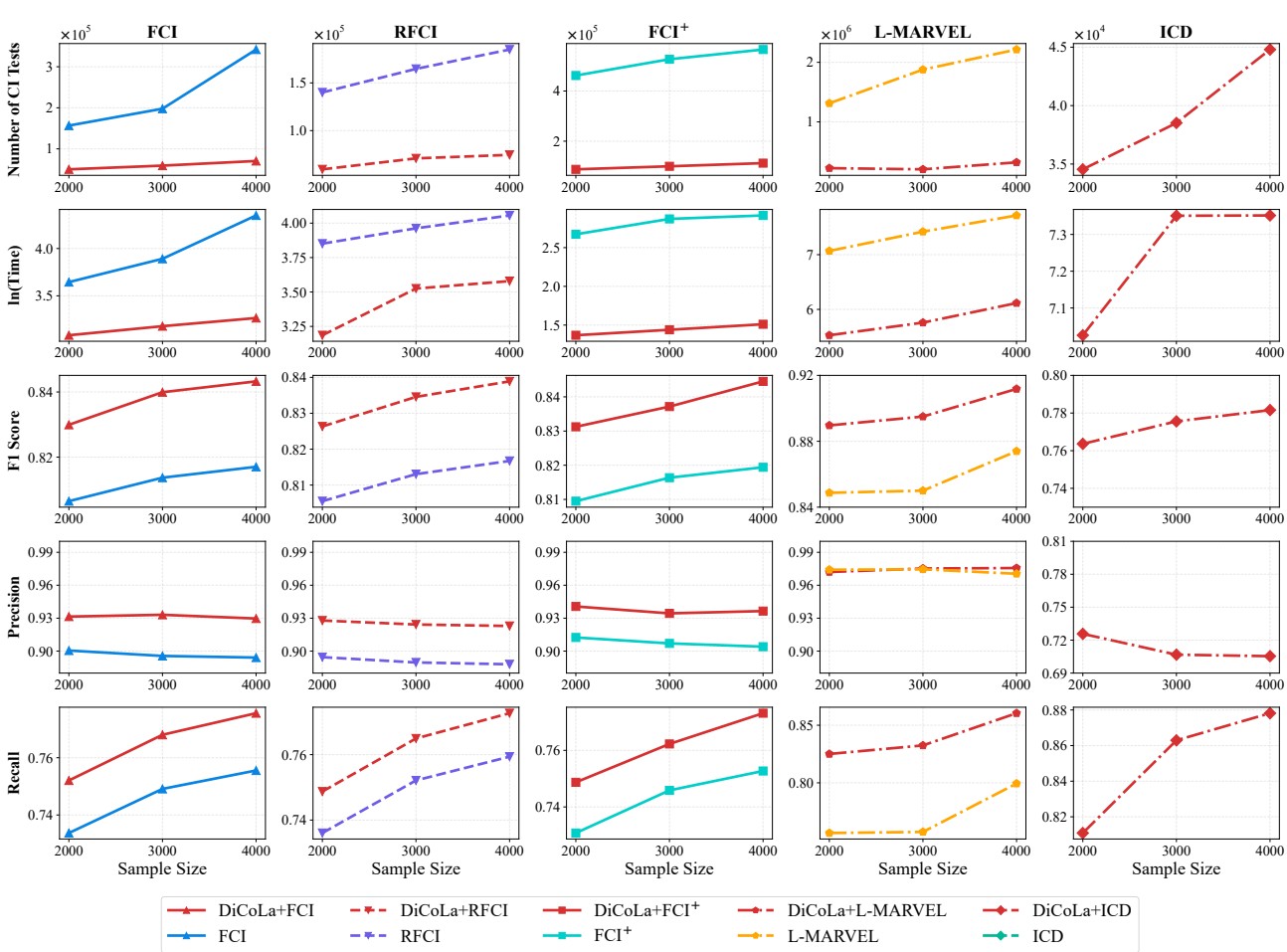

*Figure 11.* Performance on ANDES network with 10 latent variables.

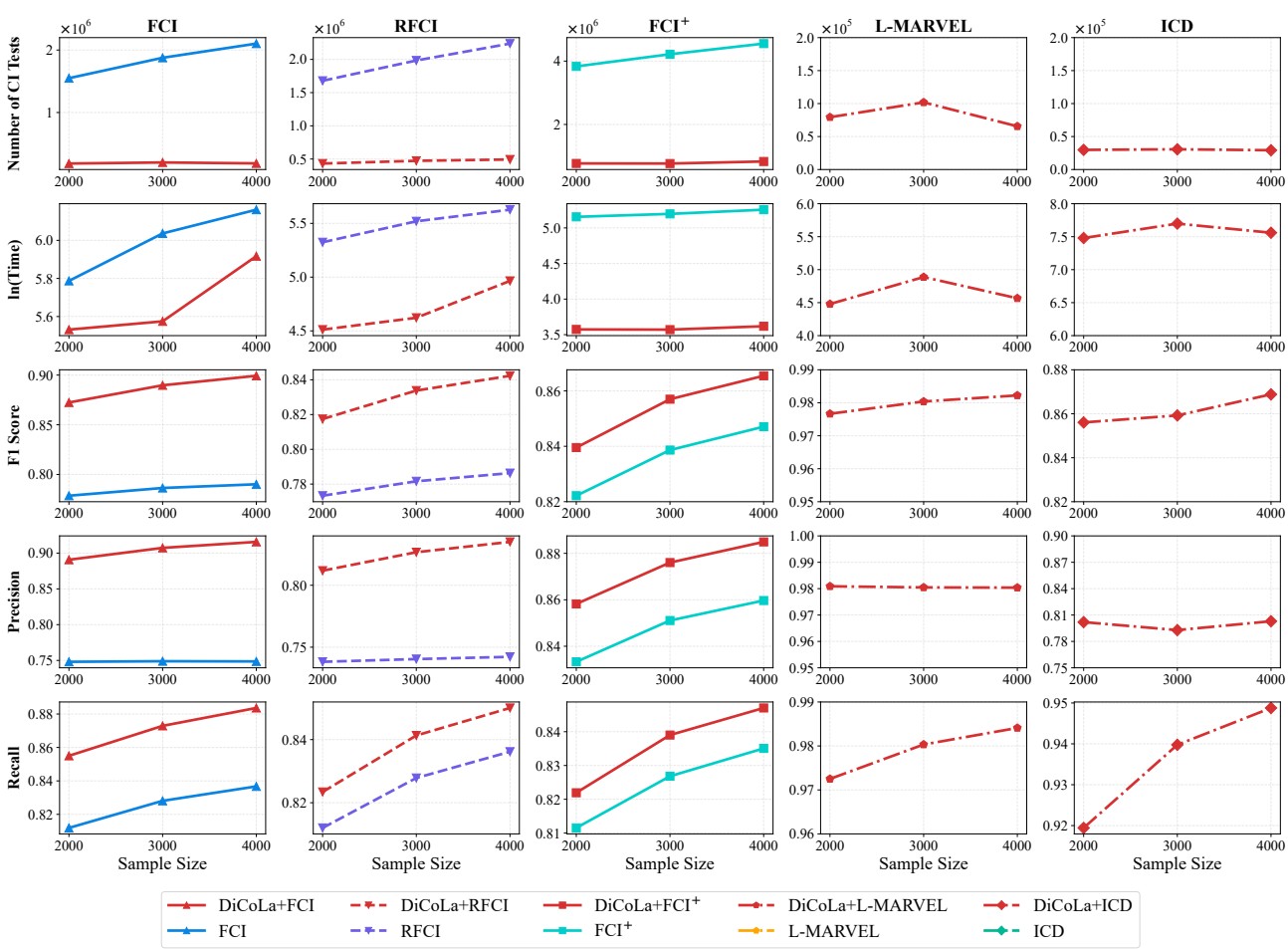

*Figure 12.* Performance on LINK network with 50 latent variables.

# D. Limitations and Discussion

We discuss the scope and limitations of DICOLA from three perspectives. First, Appendix D.1 examines its performance on dense graphs. Second, Appendix D.2 discusses its relation to local causal structure learning-based methods, which address a different task. Third, Appendix D.3 clarifies its connection to methods for learning causal models with latent variables under additional assumptions.

## D.1. Limitations under Dense Structures

The efficiency gain of DICOLA is most pronounced on sparse or well-decomposable graphs. As discussed in the complexity analysis in Section 5.2, this gain stems from reducing the size of the largest leaf subproblem, denoted by $k_{\max}$, relative to the full variable set size $n$. When useful separators exist, recursive decomposition splits the original problem into substantially smaller subproblems, leading to fewer conditional independence tests and lower runtime. In denser graphs, such separators become harder to identify, and the resulting subproblems become larger, making $k_{\max}$ closer to $n$. Thus, the computational benefit of DICOLA gradually decreases, and in the extreme case where no valid tripartition can be found, DICOLA simply falls back to the base structure learning algorithm on the full variable set.

*Table 3.* Average degrees of the induced MAGs for $\mathrm{ER}(40, d)$ graphs with 2 latent variables.

| Underlying Structure | \|V\| | \|L\| | Avg. Degree in DAGs | Avg. Degree in Induced MAGs |
|---|---|---|---|---|
| $\mathrm{ER}(40, 3)$ graphs | 40 | 2 | 3.00 | 3.33 |
| $\mathrm{ER}(40, 4)$ graphs | 40 | 2 | 4.00 | 4.83 |
| $\mathrm{ER}(40, 5)$ graphs | 40 | 2 | 5.00 | 6.20 |
| $\mathrm{ER}(40, 6)$ graphs | 40 | 2 | 6.00 | 7.83 |

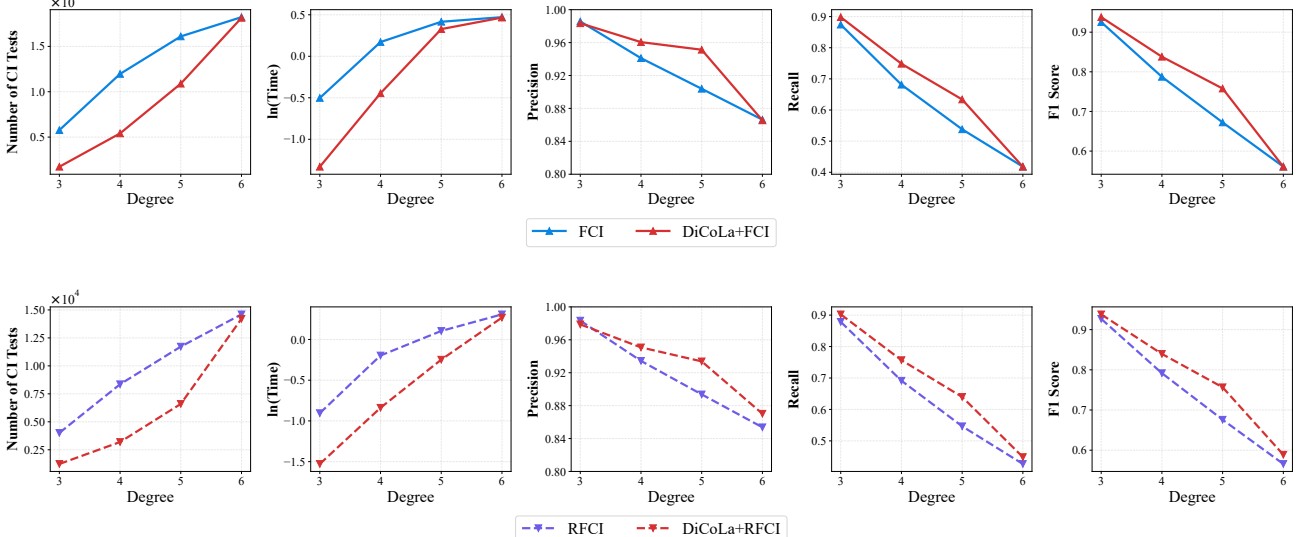

*Figure 13.* Performance on $\mathrm{ER}(40, d)$ graphs with 2 latent variables.

Importantly, this limitation affects the efficiency gain rather than the correctness of the framework. The soundness and completeness guarantees of DICOLA remain valid whenever the guarantees of the base learner hold.

To empirically examine this behavior, we conduct additional experiments using two classical constraint-based methods, FCI and RFCI, as base learners. Specifically, we evaluate them on $\mathrm{ER}(40, d)$ graphs with 2 latent variables and sample size 2000, where the average degree $d$ of the underlying DAG varies from 3 to 6. Notably, marginalizing latent variables can make the induced MAGs noticeably denser than the generating DAGs (Richardson & Spirtes, 2002); the average degrees of the induced MAGs are reported in Table 3. The results in Figure 13 show a clear trend: as the induced MAG becomes denser, the computational gain of DICOLA gradually decreases, while its structural accuracy remains comparable to that

of the corresponding base learner. For example, when using FCI as the base learner, the reduction in the number of CI tests decreases from $70\%$ to $0.4\%$ as $d$ increases from 3 to 6. Similarly, when using RFCI as the base learner, the reduction decreases from $69\%$ to $2.7\%$. These results confirm that DICOLA is most beneficial on sparse or decomposable graphs and gradually approaches the behavior of the base learner on denser graphs.

### D.2. Relation to Local Causal Structure Learning-Based Methods

We further discuss the relation between DICOLA and local causal structure learning methods in the presence of latent variables. Local methods, such as MMB-by-MMB (Xie et al., 2024b), are designed to identify the local causal structure around a given target variable, e.g., its direct causes and direct effects. In contrast, DICOLA is a recursive decomposition framework for global causal structure learning: it aims to recover the causal structure over the full observed variable set by decomposing the original problem into smaller subproblems and then integrating their outputs. Therefore, although the learned global structure can be used to extract the local structure of a target variable, DICOLA is not specifically optimized for the local discovery task.

To empirically examine this difference, we compare DICOLA+FCI with FCI and MMB-by-MMB on the task of identifying the direct causes and direct effects of a target variable. The experiments are conducted on $\mathrm{ER}(50, 3)$ graphs with 5 latent variables under different sample sizes. For each graph, we randomly choose target variables among vertices with relatively large adjacency sets. As shown in Table 4, DICOLA+FCI substantially reduces the number of conditional independence tests and runtime compared with FCI, while maintaining comparable or slightly better structural accuracy. MMB-by-MMB achieves the best performance in this local discovery task, which is expected since it is specifically designed for local causal structure learning. In contrast, DICOLA targets global structure learning and uses recursive decomposition to accelerate global constraint-based causal discovery. These results suggest that local methods are preferable when only the neighborhood of a prespecified target variable is needed, while adapting the decomposition principle of DICOLA more directly to local causal structure learning is an interesting direction for future work.

*Table 4.* Comparison with local causal structure learning on $\mathrm{ER}(50, 3)$ graphs with 5 latent variables.

| Sample Size | Algorithm | CI Tests ↓ | Time ↓ | Precision ↑ | Recall ↑ | F1 ↑ |
|---|---|---|---|---|---|---|
| | DICOLA+FCI | 2664.46 | 0.25 | 0.71 | 0.73 | 0.72 |
| 2000 | MMB-by-MMB | **909.48** | **0.07** | **0.76** | **0.77** | **0.76** |
| | FCI | 8623.34 | 0.54 | 0.70 | 0.73 | 0.71 |
| | DICOLA+FCI | 3164.88 | 0.29 | 0.75 | 0.76 | 0.75 |
| 3000 | MMB-by-MMB | **1311.88** | **0.10** | **0.77** | **0.78** | **0.77** |
| | FCI | 10089.20 | 0.62 | 0.72 | 0.73 | 0.72 |
| | DICOLA+FCI | 3421.24 | 0.30 | 0.73 | 0.79 | 0.75 |
| 4000 | MMB-by-MMB | **1763.56** | **0.14** | **0.78** | **0.79** | **0.78** |
| | FCI | 11742.28 | 0.72 | 0.71 | 0.74 | 0.72 |

Note: ↑ indicates that a higher value is better, while ↓ indicates that a lower value is better. The best result in each group is highlighted in bold.

We also note that local-graph-based acceleration methods, such as lFCI (Chen et al., 2023), are related to DICOLA in that both aim to improve the efficiency of causal structure learning in the presence of latent variables. However, the two approaches exploit different sources of efficiency. lFCI accelerates FCI by restricting the search for separating sets to local graphs induced by short paths between variable pairs. It relies on additional locality assumptions, such as the underlying graph admits small local separators and that the relevant conditional dependencies can be determined locally; for maximally informative PAG recovery, it further requires a local discriminating-path assumption. In contrast, DICOLA improves efficiency through recursive decomposition of the global problem and does not impose such local-graph assumptions on the underlying MAG.

### D.3. Relation to Learning Causal Models with Latent Variables

We further clarify the relation between DICOLA and methods for learning causal models with latent variables. DICOLA follows the standard nonparametric constraint-based setting with latent variables, where the goal is to learn the causal structure among observed variables. It does not aim to explicitly recover the latent variables themselves.

This setup differs from methods that explicitly model or identify latent structures, or causal structures involving latent variables, under additional assumptions. For example, Squires et al. (2022) focus on causal structure discovery between clusters induced by latent factors, under a parametric framework and additional structural assumptions, such as unique latent-parent clusters and a bipartite observed-latent graph. Xie et al. (2020; 2024a) consider latent variable causal discovery in linear non-Gaussian models via the generalized independent noise condition. Dong et al. (2024) further consider a rank-based framework for latent linear causal models with causally related hidden variables, relying on covariance-rank information together with assumptions such as linearity and rank faithfulness.

These methods can handle richer latent-variable structures, but their guarantees rely on specialized assumptions. In contrast, DICOLA is designed for general causal structure learning among observed variables in the presence of latent variables, without relying on these assumptions. Combining recursive decomposition with additional latent-variable modeling assumptions is an interesting direction for future work.

