# OpenReview forum: "A Recursive Decomposition Framework for Causal Structure Learning in the Presence of Latent Variables"
_ICML.cc/2026/Conference — ICML 2026 spotlight_

### Official Review · Reviewer_etge · 2026-02-14

**Soundness:** 3
**Presentation:** 3
**Significance:** 3
**Originality:** 3
**Overall Recommendation:** 5
**Confidence:** 3

**Summary:**

This paper provides a theorectial foundation for a recursive decomposition method for causal networks learning when latent variables are present. Emperical evaluation using simulation and real datasets show strong performance and improve efficiency over methods without decomposition

**Compliance With Llm Reviewing Policy:**

Affirmed.

**Final Justification:**

The author’s rebuttal addresses all my concerns

**Key Questions For Authors:**

1. It would be great if the authors can discuss the limitations of the algorithm, e.g., what if a latent variable is connected to many observed variables.

**Limitations:**

Yes

**Strengths And Weaknesses:**

This paper is well written and well structure with sound theoretical results and solid experimental data. It provides a good tool for learning causal network structures from high dimensional sparse datasets.

It would be great if the authors could compare this method with other divide-and-conquer approches such as clustering base methods (e.g., "Causal Structure Discovery between Clusters of Nodes Induced by Latent Factors" by Squires et al.), and local structrue learning based methods ( "Local Causal Structure Learning in the Presence of Latent Variables" by Xie et al. and "Causal Structural Learning Via Local Graphs" by Chen et al.?)

---

> ### Author Rebuttal · Authors · 2026-03-29
>
> We sincerely thank you for recognizing our theoretical soundness, solid empirical performance, and practicality. We hope the following responses properly address your concerns.
>
> >**W1:** It would be great if the authors could compare this method with other divide-and-conquer approches such as clustering base methods (e.g., Squires et al.), and local structrue learning based methods ( Xie et al. and Chen et al.?)
>
> **A1:** Thank you for this suggestion.
>
> **1. Squires et al. :** We would like to clarify that they address a different problem setting. Specifically, they focus on causal structure discovery between clusters induced by latent factors under a parametric framework and additional structural assumptions on the latent variables. In contrast, DiCoLa, like standard constraint-based methods such as FCI and RFCI, aims to learn the causal structure among observed variables in a nonparametric setting. We will clarify this distinction in the revised manuscript.
>
> **2. Xie et al. :** This proposes MMB-by-MMB for *local* causal structure learning in the presence of latent variables. We added an empirical comparison on this task, i.e., identifying the direct causes and effects of a target variable. Specifically, we compare DiCoLa+FCI with FCI and MMB-by-MMB. As shown below, while DiCoLa+FCI significantly outperforms FCI in efficiency, while also improving structural accuracy, MMB-by-MMB naturally performs best here. This is expected, as MMB-by-MMB is specifically designed for local causal structure learning, whereas DiCoLa is designed for *global* causal structure learning. Adapting DiCoLa for local tasks is an interesting direction for future work.
>
> |Network|Size|Algorithm|CI Tests↓|Time↓|Prec↑|Rec↑|F1↑|
> |-|-|-|-|-|-|-|-|
> |**ER(50,3) 5 latents**|2000|DiCoLa+FCI|2664.46|0.25|0.71|0.73|0.72|
> |||MMB-by-MMB|909.48|0.07|0.76|0.77|0.76|
> |||FCI|8623.34|0.54|0.70|0.73|0.71|
> ||3000|DiCoLa+FCI|3164.88|0.29|0.75|0.76|0.75|
> |||MMB-by-MMB|1311.88|0.10|0.77|0.78|0.77|
> |||FCI|10089.20|0.62|0.72|0.73|0.72|
> ||4000|DiCoLa+FCI|3421.24|0.30|0.73|0.79|0.75|
> |||MMB-by-MMB|1763.56|0.14|0.78|0.79|0.78|
> |||FCI|11742.28|0.72|0.71|0.74|0.72|
>
> **3. Chen et al. :** They propose local FCI (lFCI), accelerating FCI by restricting separating-set searches to local graphs induced by short paths. However, lFCI relies on additional locality assumptions, namely that the underlying graph admits small local separators and that the relevant conditional dependencies can be determined locally; for maximally informative PAG recovery, it further requires a local discriminating-path assumption. In contrast, DiCoLa does not rely on this type of local-graph assumption on the underlying MAG. We did not include an empirical comparison with lFCI in the current rebuttal because we were unable to find an official public implementation. We will continue investigating this comparison and, if feasible, include it in the revised manuscript.
>
>
>
> >**Q2:** It would be great if the authors can discuss the limitations of the algorithm, e.g., what if a latent variable is connected to many observed variables.
>
> **A2:** When a latent variable is connected to many observed variables, the induced MAG typically becomes denser, which makes valid decomposition harder to find and can reduce the computational gain of DiCoLa. More broadly, under the nonparametric constraint-based setting considered in our paper, such strong latent confounding also makes causal structure learning itself more difficult.
>
> We note, however, that related scenarios can sometimes be handled under additional assumptions. For example, Squires et al. study a parametric latent-factor model with certain structural assumptions such as unique latent-parent clusters and a bipartite observed-latent graph. Xie et al. [1,2] consider latent variable causal discovery in linear non-Gaussian models via the generalized independent noise condition. Dong et al. [3] further consider a rank-based framework for latent linear causal models with causally related hidden variables, relying on covariance-rank information together with assumptions such as linearity and rank faithfulness. In contrast, DiCoLa is designed for general causal structure learning among observed variables in the presence of latent variables, without relying on these specialized assumptions. We will add this limitation and discussion to the revised manuscript.
>
> [1] Xie, F., Cai, R., Huang, B., Glymour, C., Hao, Z. and Zhang, K. Generalized independent noise condition for estimating latent variable causal graphs. NeurIPS 2020.
>
> [2] Xie, F., Huang, B., Chen, Z., Cai, R., Glymour, C., Geng, Z., and Zhang, K. Generalized independent noise condition for estimating causal structure with latent variables. JMLR, 2024.
>
> [3] Dong, X., Huang, B., Ng, I., Song, X., Zheng, Y., Jin, S., Legaspi R., Spirtes P., and Zhang, K. A Versatile Causal Discovery Framework to Allow Causally-Related Hidden Variables. ICLR 2024.

---

> > ### Author Rebuttal · Reviewer_etge · 2026-04-02
> >
> > Thanks for addressing my concerns and providing additioanl experimental results.

---

> > > ### Author Response · Authors · 2026-04-03
> > >
> > > Thank you very much for your positive feedback and for acknowledging that your concerns have been fully addressed!

---

### Official Review · Reviewer_4nHw · 2026-03-05

**Soundness:** 3
**Presentation:** 4
**Significance:** 3
**Originality:** 3
**Overall Recommendation:** 5
**Confidence:** 3

**Summary:**

This paper addresses the poor scalability of constraint-based methods in the presence of latent variables. The authors introduce DICOLA, a novel recursive divide-and-conquer framework. By utilizing Markov blankets to construct Undirected Independence Graphs (UIGs), the framework decomposes the global learning task into smaller, overlapping subproblems. Base algorithms (such as FCI or RFCI) are then run on these subproblems, and their local skeletons are merged to reconstruct the global PAG. The authors provide theoretical proofs for the soundness and completeness of this framework and demonstrate substantial empirical improvements in computational efficiency across several benchmark datasets.

**Compliance With Llm Reviewing Policy:**

Affirmed.

**Final Justification:**

The author’s rebuttal addresses my minor concerns and I’d like to maintain my positive score.

**Key Questions For Authors:**

I have no questions.

**Limitations:**

yes

**Strengths And Weaknesses:**

Overall, I think this is a good paper.

## Strengths

1. Extending the divide-and-conquer paradigm from causally sufficient settings to latent variable settings is promising. Theorems 1 to 3 are well-formulated and nicely bridge the gap between global m-separations and local subproblem separations.

2. DICOLA is a plug-and-play module that accelerates diverse methods like FCI, RFCI, L-MARVEL, and ICD, which is a major practical advantage.

3. The authors provide a very thorough empirical evaluation.

## Weaknesses

I only have two minor concerns.

1. The framework's success hinges entirely on the accurate construction of the Undirected Independence Graph (UIG) via Markov Blanket (MB) discovery during Phase 1. In high-dimensional or relatively dense causal structures, standard MB discovery algorithms (such as the TC algorithm used in the experiments) suffer from decreased reliability.

2. While the paper compares against standard FCI, RFCI, L-MARVEL, and ICD, it omits FCI+ (Claassen et al., 2013), which is a very standard, highly optimized variant of FCI designed specifically to reduce CI tests.

---

> ### Author Rebuttal · Authors · 2026-03-29
>
> We sincerely thank you for the constructive and insightful feedback, and for recognizing both the theoretical value of our framework and the practical ``plug-and-play'' nature of DiCoLa. We hope that the following responses adequately address your concerns.
>
>
> >**W1:** The framework's success hinges entirely on the accurate construction of the UIG via MB discovery during Phase 1.  In high-dimensional or relatively dense causal structures, standard MB discovery algorithms (such as the TC ) suffer from decreased reliability.
>
> **A1:** We agree that the quality of Phase 1 depends on the reliability of MB discovery, and that this can become more challenging in high-dimensional or relatively dense settings. In our experiments, we followed [1] and used the TC algorithm in order to ensure a fair comparison with prior work.
>
> At the same time, DiCoLa is not tied to a specific MB learner. It can also be combined with more recent MB discovery methods designed for high-dimensional settings, such as STMB [2], EAMB [3], and CFS-MI [4]. More broadly, improving MB discovery under latent confounding, or developing decomposition procedures that do not rely on MB learning, are both important directions for future work. We will clarify this accordingly.
>
> [1] Mokhtarian E, et al. Recursive causal discovery. JMLR, 2025.
>
> [2] Gao T, Ji Q. Efficient Markov blanket discovery and its application. IEEE transactions on Cybernetics, 2016.
>
> [3] Guo X,  et al. Error-aware Markov blanket learning for causal feature selection. Information Sciences, 2022.
>
> [4] Ling Z, et al.A light causal feature selection approach to high-dimensional data. IEEE TKDE, 2022.
>
> >**W2:** While the paper compares against standard FCI, RFCI, L-MARVEL, and ICD, it omits FCI+ (Claassen et al., 2013), which is a very standard, highly optimized variant of FCI designed specifically to reduce CI tests.
>
> **A2:** We have added a performance comparison with FCI$^{+}$ under the same settings as in our paper. As shown in the table below, DiCoLa+FCI$^{+}$ consistently reduces the number of CI tests and runtime relative to FCI$^{+}$, while matching or improving structural accuracy across most settings.
>
> |Network|Size|Alg.|CI Tests↓|Time↓|Pre↑|Rec↑|F1↑|
> |-|-|-|-|-|-|-|-|
> |**MILDEW**|2000|DiCoLa+FCI$^{+}$|4705.14|0.28|0.95|0.86|0.91|
> |||FCI$^{+}$|9585.00|0.69|0.93|0.84|0.89|
> ||3000|DiCoLa+FCI$^{+}$|5070.62|0.27|0.95|0.87|0.91|
> |||FCI$^{+}$|10460.42|0.72|0.94|0.85|0.89|
> ||4000|DiCoLa+FCI$^{+}$|5434.32|0.28|0.95|0.88|0.92|
> |||FCI$^{+}$|11288.12|0.75|0.94|0.86|0.90|
> |**BARLEY**|2000|DiCoLa+FCI$^{+}$|14729.38|0.51|0.89|0.64|0.75|
> |||FCI$^{+}$|23415.82|1.44|0.89|0.64|0.74|
> ||3000|DiCoLa+FCI$^{+}$|15778.70|0.51|0.89|0.66|0.76|
> |||FCI$^{+}$|26865.42|1.53|0.89|0.65|0.75|
> ||4000|DiCoLa+FCI$^{+}$|16409.78|0.53|0.89|0.67|0.76|
> |||FCI$^{+}$|29589.32|2.01|0.88|0.66|0.75|
> |**ANDES**|2000|DiCoLa+FCI$^{+}$|87850.24|3.93|0.94|0.75|0.83|
> |||FCI$^{+}$|461714.74|14.47|0.91|0.73|0.81|
> ||3000|DiCoLa+FCI$^{+}$|99967.12|4.22|0.93|0.76|0.84|
> |||FCI$^{+}$|526601.20|17.65|0.91|0.75|0.82|
> ||4000|DiCoLa+FCI$^{+}$|112517.10|4.53|0.94|0.77|0.84|
> |||FCI$^{+}$|565299.96|18.46|0.90|0.75|0.82|
> |**LINK**|2000|DiCoLa+FCI$^{+}$|769762.16|35.63|0.86|0.82|0.84|
> |||FCI$^{+}$|3832296.96|173.42|0.83|0.81|0.82|
> ||3000|DiCoLa+FCI$^{+}$|766377.44|35.54|0.88|0.84|0.86|
> |||FCI$^{+}$|4216180.80|180.80|0.85|0.83|0.84|
> ||4000|DiCoLa+FCI$^{+}$|830971.84|37.26|0.88|0.85|0.87|
> |||FCI$^{+}$|4551417.14|191.76|0.86|0.84|0.85|
> |**ER(30,3) with 3 latent variables**|2000|DiCoLa+FCI$^{+}$|4363.92|0.34|0.98|0.78|0.86|
> |||FCI$^{+}$|5665.26|0.46|0.98|0.77|0.86|
> ||3000|DiCoLa+FCI$^{+}$|4635.30|0.40|0.98|0.80|0.88|
> |||FCI$^{+}$|5941.66|0.61|0.98|0.80|0.88|
> ||4000|DiCoLa+FCI$^{+}$|5363.30|0.45|0.98|0.81|0.89|
> |||FCI$^{+}$|6562.58|0.67|0.98|0.81|0.88|
> |**ER(30,5) with 3 latent variables**|2000|DiCoLa+FCI$^{+}$|10440.96|0.65|0.92|0.45|0.60|
> |||FCI$^{+}$|11451.76|0.76|0.91|0.44|0.59|
> ||3000|DiCoLa+FCI$^{+}$|12170.98|0.56|0.91|0.46|0.61|
> |||FCI$^{+}$|13223.80|0.86|0.90|0.45|0.60|
> ||4000|DiCoLa+FCI$^{+}$|13532.18|0.79|0.91|0.46|0.61|
> |||FCI$^{+}$|14351.34|0.93|0.91|0.45|0.60|
> |**ER(50,3) with 5 latent variables**|2000|DiCoLa+FCI$^{+}$|7292.62|0.56|0.98|0.80|0.88|
> |||FCI$^{+}$|15273.60|1.06|0.98|0.80|0.88|
> ||3000|DiCoLa+FCI$^{+}$|7988.82|0.59|0.98|0.83|0.90|
> |||FCI$^{+}$|16371.50|1.14|0.98|0.82|0.89|
> ||4000|DiCoLa+FCI$^{+}$|8736.28|0.61|0.98|0.84|0.90|
> |||FCI$^{+}$|17654.24|1.24|0.98|0.82|0.89|
> |**ER(50,3) with 7 latent variables**|2000|DiCoLa+FCI$^{+}$|4749.10|0.91|0.98|0.76|0.85|
> |||FCI$^{+}$|15645.30|1.42|0.98|0.75|0.85|
> ||3000|DiCoLa+FCI$^{+}$|5611.82|0.86|0.98|0.79|0.87|
> |||FCI$^{+}$|17547.82|1.70|0.99|0.78|0.87|
> ||4000|DiCoLa+FCI$^{+}$|6572.16|0.86|0.99|0.80|0.88|
> |||FCI$^{+}$|19004.26|1.67|0.99|0.79|0.87|
>
> Overall, these additional results further support the main claim of the paper. DiCoLa remains effective even when combined with an already optimized CI-based learner such as FCI$^{+}$. We will include these results in the revised manuscript.

---

> > ### Author Rebuttal · Reviewer_4nHw · 2026-04-01
> >
> > Thanks for your rebuttal. I'd like to maintain my positive score.

---

> > > ### Author Response · Authors · 2026-04-02
> > >
> > > Thank you very much for your positive feedback and appreciation of our work!

---

### Official Review · Reviewer_b3n3 · 2026-03-09

**Soundness:** 3
**Presentation:** 3
**Significance:** 3
**Originality:** 3
**Overall Recommendation:** 4
**Confidence:** 4

**Summary:**

The authors proposed a divide-and-conquer method for constraint-based causal discovery in the presence of latent variables, with soundness and completeness guarantees. The method was shown to be computationally more tractable while maintaining statistical accuracy.

**Compliance With Llm Reviewing Policy:**

Affirmed.

**Final Justification:**

My questions are addressed, and I maintain my positive score.

**Key Questions For Authors:**

1. What is the main technical novelty in generalizing the divide-and-conquer strategy from the setting without latent confounders?
2. How does the proposed method relate to the iterative and recursive techniques mentioned and compared by the authors?
3. When constructing the undirected independence graph, it is likely that some conditional independences in the original maximal ancestral graph are lost. It is therefore unclear how this may impact the discovery of the tripartition, and why the authors propose introducing the undirected independence graph in the first place. Perhaps this is mainly for computational reasons, but the rationale should be clarified.
4. The divide-and-conquer strategy is mainly used for recovering the skeleton, while the orientation step is still performed globally. Is this a common practice, or is it impossible to perform orientation locally? Alternatively, does performing orientation globally not impact the computational cost?
5. While the method reduces the number of conditional independence tests, it may still require large conditioning sets in the tripartition recovery step. Moreover, the extent of the reduction in the number of required conditional independence tests depends on the underlying graph structure. This point is somewhat demonstrated in the simulation studies, but I wonder whether it is possible to establish a more formal conclusion, perhaps for a restricted class of graphs.

**Limitations:**

The limitations of the method are not discussed. The authors may consider describing scenarios in which the proposed approach could be outperformed by methods that do not use a divide-and-conquer strategy, particularly in terms of accuracy.

**Strengths And Weaknesses:**

The problem is well motivated, and the presentation is clear. As claimed by the authors, this is the first method that considers a divide-and-conquer strategy for constraint-based causal discovery in the presence of latent confounders, which could have a broad range of applications. I did not check all technical details, but the theoretical results appear to be sound. See the questions below for potential weaknesses.

---

> ### Author Rebuttal · Authors · 2026-03-29
>
> We sincerely thank the reviewer for the thoughtful comments and for recognizing the motivation, clarity, soundness, and significance of our work. Our responses are as follows.
>
> > **Q1:** … technical novelty … from the setting without latent confounders?
>
> **A1:** The main technical novelty is that, with latent confounders, the relevant graph is a MAG rather than a DAG, so the standard d-separation arguments used in prior divide-and-conquer methods no longer apply. Instead, the correct separation notion is m-separation, and bidirected edges induced by latent confounders can introduce dependencies across different parts of the graph. Thus, it is no longer immediate that a valid decomposition preserves the m-separation information needed for correct local learning and global reconstruction. We show that divide-and-conquer remains valid: Theorems 1–2 localize global m-separation to subproblems, Theorem 3 ensures correct skeleton reconstruction, and we provide an MB-based procedure to find valid decompositions from data.
>
> >**Q2:** … relation to iterative/recursive techniques …
>
> **A2:**  We would like to clarify that DiCoLa is a divide-and-conquer framework, rather than a standalone causal discovery algorithm. Methods such as ICD and L-MARVEL are standalone causal discovery algorithms: ICD reduces CI testing by gradually enlarging graph-distance-based conditioning sets, while L-MARVEL does so through recursive elimination of a specific class of variables. Thus, they operate at different levels: ICD and L-MARVEL are base learners, whereas DiCoLa is a framework that can be combined with them.
>
> >**Q3:** … UIG may lose conditional independences (CIs) in the MAG … impact on tripartition … rationale …
>
> **A3:** The primary reason for introducing the UIG is computational. It provides an efficient surrogate structure for finding a valid tripartition $(A,B,C)$, without first reconstructing the full MAG.
>
> Importantly, the UIG does not need to preserve all CIs in the MAG. It is sufficient that any separation found in the UIG implies a valid m-separation in the MAG (Definition 1). Therefore, missing CIs may reduce the number of decompositions, but do not affect the soundness of any tripartition returned by Algorithm 2.
>
> >**Q4:**  … global orientation … local vs global … cost …
>
> **A4:** Following prior divide-and-conquer work (e.g., Xie and Geng, 2008), we perform global orientation after reconstructing the skeleton. This is because orientation is not purely local, and some orientation rules depend on paths or configurations that may only become visible after subproblems are merged. For example, even if one first orients the local skeletons of leaf subproblems (e.g., $\mathcal{L}_6,\mathcal{L}_7$ in Figure 2(e)), additional orientation would still be needed after intermediate merging (e.g., $\mathcal{L}_3$ in Figure 2(e)), and a final global orientation step would still be required to ensure completeness. Moreover, in constraint-based causal discovery, the main computational bottleneck lies in CI testing during skeleton discovery rather than in orientation. Therefore, keeping the orientation step global does not affect DiCoLa’s primary efficiency gain from reducing CI tests.
>
> >**Q5:** Fewer CI tests but may require large conditioning sets in the tripartition recovery step … underlying structure-dependent gains … formal …
>
> **A5:**  We agree that large conditioning sets may arise in MB learning for tripartition recovery. We used the TC algorithm in order to ensure a fair comparison with prior work. DiCoLa is also compatible with alternative MB discovery methods that are designed to work with smaller conditioning sets, such as MMMB [1] and HITON-MB [2].
>
> We also agree that the benefit of DiCoLa depends on the decomposability of the underlying graph. A formal characterization can be stated in terms of the largest leaf subproblem size $k_{max}$. As shown by our complexity analysis, the computational gain is substantial when the recursive decomposition yields $k_{max} \ll n$, and it correspondingly diminishes when the graph is poorly decomposable and $k_{max}$ approaches $n$. This is expected, for example, for sparse graphs that admit small separators throughout the decomposition process.
>
> [1] Tsamardinos I, et al. Time and sample efficient discovery of Markov blankets and direct causal relations. KDD, 2003.
>
> [2] Aliferis C F, et al. HITON: a novel Markov Blanket algorithm for optimal variable selection. 2003.
>
> > **Limitations** … where non–divide-and-conquer methods may outperform …
>
> **A6:**  A main limitation of DiCoLa is that it is most effective on sparse or decomposable graphs. For dense graphs or those lacking useful separators, it may offer limited computational gains and reduce to the base learner. In addition, its finite-sample performance depends on the quality of MB discovery in Phase 1; unreliable MB estimation may lead to less reliable decomposition and affect downstream accuracy. We will add this discussion.

---

> > ### Author Rebuttal · Reviewer_b3n3 · 2026-04-02
> >
> > Thank you for the response. I will maintain my score.

---

> > > ### Author Response · Authors · 2026-04-03
> > >
> > > Thank you for the reconsideration and for confirming that the concerns were fully resolved. We appreciate the positive assessment!

---

### Official Review · Reviewer_WtQq · 2026-03-12

**Soundness:** 3
**Presentation:** 3
**Significance:** 2
**Originality:** 3
**Overall Recommendation:** 4
**Confidence:** 4

**Summary:**

This paper proposes a method for learning causal structures in high-dimensional data and scenarios containing latent variables. The proposed method employs a divide-and-conquer approach, dividing the variables into smaller sets, learning the causal structure within each set, and finally merging the results. Detailed theoretical explanations and proofs are provided, and experiments validate the effectiveness of the proposed method.

**Compliance With Llm Reviewing Policy:**

Affirmed.

**Final Justification:**

The author provided a detailed response that addressed my concerns, but it wasn't enough to improve the score.

**Key Questions For Authors:**

See above.

**Limitations:**

See above.

**Strengths And Weaknesses:**

Strengths:
1. For large-scale data scenarios, the divide-and-conquer approach is a classic strategy. This paper provides relevant theoretical explanations and proofs, clearly demonstrating that point sets can be partitioned under different conditions, thus enabling the learning of causal structures as the learning of local structures without compromising identifiability.
2. For high-dimensional data scenarios, constraint-based causal discovery methods typically require numerous conditional independence tests, and the condition sets are often large, increasing algorithm time consumption and reducing stability. The divide-and-conquer approach reduces these issues, significantly lowering the computational cost of conditional independence tests and improving their reliability.
3. Experiments have verified that the proposed method can effectively reduce the number of conditional tests, reduce algorithm time, and improve the learning accuracy of causal structures.

Weaknesses:
1. The proposed method is effective for sparse causal structures, but its performance easily degrades to that of base structure learning algorithms for dense graphs.
2. In the synthetic data experiments, the random graphs used were still relatively sparse. Using denser graphs to verify the performance of the proposed method on dense graphs would be more convincing.

---

> ### Author Rebuttal · Authors · 2026-03-29
>
> We sincerely thank you for the constructive and thoughtful feedback. We also appreciate your positive assessment of the motivation, theoretical results, and empirical evaluation. We hope the following responses address the concerns clearly.
>
> >**W1** ``The proposed method is effective for sparse causal structures, but its performance easily degrades to that of base structure learning algorithms for dense graphs.''
>
>
> **A1:** Thank you for this important observation. We agree that the efficiency gain of DiCoLa is most evident on sparse or well-decomposable graphs. Similar to many scalable constraint-based methods, its computational advantage becomes smaller as the graph becomes denser.
>
> This is also consistent with our complexity analysis. The benefit of DiCoLa comes from reducing the size of the largest leaf subproblem $k_{max}$ compared with the full variable set size $n$. When the graph becomes denser, useful separators are harder to find, the subproblems become larger, and $k_{max}$ gets closer to $n$. As a result, the benefit of recursive decomposition gradually decreases. In the extreme case where no valid tripartition can be found, DiCoLa simply falls back to the base structure learning algorithm on the full variable set.
>
>
> We would like to emphasize that this limitation affects the amount of computational gain, rather than the correctness of the framework. Our main contribution is to provide the first theoretically justified divide-and-conquer framework for causal structure learning in the presence of latent variables, with soundness and completeness inherited from the base learner. We will clarify this point in the revised manuscript.
>
> >**W2** ``In the synthetic data experiments, the random graphs used were still relatively sparse. Using denser graphs to verify the performance of the proposed method on dense graphs would be more convincing.''
>
> **A2:** Thank you for this helpful suggestion. We would first like to clarify that, in our original synthetic settings, marginalizing latent variables already makes the induced MAGs noticeably denser than the generating DAGs [1]. For example, the average degree increases from 5.00 to 7.28 for ER(30,5) with 3 latent variables. In particular, following the experimental setting in [2], the induced MAGs for ER(30,5) with 3 latent variables and ER(50,3) with 7 latent variables can already be viewed as relatively dense compared with their generating DAGs. The corresponding graph settings and average degrees are summarized in the table below.
>
> | **Graph** | \|**V**\| | \|**L**\| | **Avg. Deg. (Underlying DAG)** | **Avg. Deg. (Induced MAG)** | **Reference** |
> |-|-|-|-|-|-|
> |ER(30,3) |30|3|3.00|3.81| Fig. 5|
> |ER(30,5) |30|3|5.00|7.28| Fig. 6|
> |ER(50,3) |50|5|3.00|3.83| Fig. 7|
> |ER(50,3) |50|7|3.00|4.31| Fig. 8|
>
>
> Following your suggestion, we further conducted additional experiments on ER(40,$d$) graphs with 2 latent variables and sample size 2000. Here, the average degree $d$ of the underlying DAG varies from 3 to 6, allowing us to more directly examine the dense-graph setting. The results are shown in the table below.
>
> |Avg. Deg. (Underlying DAG)|Avg. Deg. (Induced MAG)|Algorithm|No. of CI Tests↓|Time↓|Precision↑|Recall↑|F1↑|
> |-|-|-|-|-|-|-|-|
> |3|3.33|DiCoLa+FCI|1727.22|0.26|0.98|0.90|0.94|
> |||FCI|5767.52|0.61|0.99|0.87|0.93|
> |||DiCoLa+RFCI|1232.30|0.22|0.98|0.90|0.94|
> |||RFCI|4007.08|0.40|0.98|0.88|0.93|
> |4|4.83|DiCoLa+FCI|5406.18|0.64|0.96|0.75|0.84|
> |||FCI|11965.02|1.19|0.94|0.68|0.79|
> |||DiCoLa+RFCI|3193.86|0.43|0.95|0.76|0.84|
> |||RFCI|8356.18|0.82|0.93|0.69|0.79|
> |5|6.20|DiCoLa+FCI|10888.56|1.39|0.95|0.63|0.76|
> |||FCI|16101.50|1.52|0.90|0.54|0.67|
> |||DiCoLa+RFCI|6597.84|0.78|0.93|0.64|0.76|
> |||RFCI|11720.60|1.11|0.89|0.55|0.68|
> |6|7.83|DiCoLa+FCI|18149.22|1.59|0.87|0.42|0.56|
> |||FCI|18224.88|1.60|0.87|0.42|0.56|
> |||DiCoLa+RFCI|14208.58|1.31|0.87|0.45|0.59|
> |||RFCI|14602.88|1.36|0.85|0.43|0.57|
>
>
> These additional results show a clear trend: as the induced MAG becomes denser, the computational gain of DiCoLa gradually decreases, while the structural accuracy remains comparable to that of the corresponding base methods. For example, the reduction in the number of CI tests drops from 70% to 0.4% for FCI and from 69% to 2.7% for RFCI as $d$ increases from 3 to 6. Overall, these results confirm that DiCoLa is most beneficial on sparse or decomposable graphs and gradually approaches the behavior of the base learner on denser graphs. We will add these results and discussions in the revised version.
>
>
> [1] Richardson T, Spirtes P. Ancestral graph Markov models. The Annals of Statistics, 2002.
>
> [2] Mokhtarian E, Elahi S, Akbari S, Kiyavash N. Recursive causal discovery. JMLR, 2025.

---

> > ### Author Rebuttal · Reviewer_WtQq · 2026-04-03
> >
> > Thanks for your detailed response. I'd like to maintain my score.

---

> > > ### Author Response · Authors · 2026-04-04
> > >
> > > Thank you for the feedback and for acknowledging that the concerns have been fully addressed!

---

### Decision · Program_Chairs · 2026-04-30

**Decision:**

Accept (spotlight)

**Comment:**

This paper proposes a divide-and-conquer method for causal structure learning in data with latent variables. Theoretical results are presented to support the graph decomposition strategies. Extensive experiments on synthetic datasets demonstrate the efficiency of the proposed method.

This is the first divide-and-conquer approach for causal structure learning in the presence of latent variables. All reviewers support accepting this paper. The theoretical results supporting the graph decomposition strategies are sound, and the efficiency improvements over benchmark methods are significant. The algorithm addresses strong practical needs in causal discovery and inference.

The authors should update the paper to incorporate the changes promised during the rebuttal process.